

# Concept, absolute calibration and validation of a new, bench-top laser imaging polar nephelometer

Alireza Moallemi[1], Robin L. Modini[1], Benjamin T. Brem[1], Barbara Bertozzi[1], Philippe Giaccari[2], and Martin Gysel-Beer[1]

[1]Laboratory of Atmospheric Chemistry, Paul Scherrer Institute, Villigen PSI, 5232, Switzerland
[2]Micos Engineering GmbH, Dübendorf, CH-8600, Switzerland

*Correspondence to*: Martin Gysel-Beer (martin.gysel@psi.ch)

**Abstract.**

Polar nephelometers provide in situ measurements of aerosol angular light scattering and play an essential role in validating numerically calculated phase functions or inversion algorithms used in space-borne and land-based aerosol remote sensing. In this study, we present a prototype of a new polar nephelometer called uNeph. The instrument is designed to measure the phase function, $F_{11}$, and polarized phase function, $-F_{12}/F_{11}$ over the scattering range of around 5˚ to 175˚ with an angular resolution of 1˚ at a wavelength of 532 nm. In this work, we present details of the data processing procedures and instrument calibration approaches. The uNeph was validated in a laboratory setting using mono-disperse polystyrene latex (PSL) and Di-Ethyl-Hexyl-Sebacate (DEHS) aerosol particles over a variety of sizes, ranging from 200 nm to 800 nm. An error model was developed and the level of agreement between uNeph measurements and Mie theory was found to be consistent within the uncertainties of the measurements and the uncertainties of the input parameters for the theoretical calculations. The estimated measurement errors were between 5% to 10 % (relative) for $F_{11}$ and smaller than ~0.1 (absolute) for $-F_{12}/F_{11}$. Additionally, by applying the Generalized Retrieval of Aerosol and Surface Properties (GRASP) inversion algorithm to the measurements conducted with broad unimodal DEHS aerosol particles, the volume concentration, size distribution and refractive index of the ensemble of aerosol particles were accurately retrieved. This paper demonstrates that the uNeph prototype can be used to conduct accurate measurements of aerosol phase function and polarized phase function and to retrieve aerosol properties through inversion algorithms.

## 1 Introduction

Atmospheric aerosol particles have a substantial impact on the Earth's radiative budget through their direct interaction with solar radiation and by affecting cloud formation processes (Boucher et al., 2013). Furthermore, aerosol particles are a significant component of air pollution which has been estimated to cause 4.2 million annual premature deaths worldwide (Cohen et al., 2017). The complex compositional, microphysical properties and high spatio-temporal variability of atmospheric aerosols makes it difficult to properly characterize and constrain them in atmospheric global model simulations leading to large uncertainties in estimation of aerosol radiative forcing contribution compared to other prevalent climate change drivers



such as $CO_2$ (Myhre et al., 2013). Global scale long term measurements of various aerosol properties are essential for obtaining in-depth insights on atmospheric aerosol variability and to develop more realistic parametrization of aerosol particles in global atmospheric models. Passive satellite and ground-based remote sensing, which are the main approaches to obtain atmospheric

aerosol observations on a global scale, rely on measurements of elastically-scattered solar radiation by atmospheric aerosols (Boucher, 2015b). Remote sensing instruments are typically designed to detect irradiance of scattered light over single or multiple angles (radiometric measurements), while certain instruments can provide complementary measurements on the scattered light polarization state (polarimetric measurements) (Dubovik et al., 2019). The polarimetric and radiometric measurements contain implicit information of numerous aerosol micro-physical properties such are refractive index (RI), size

and shape (Bohren & Huffman, 2004).

In remote sensing, aerosol properties are inferred from radiometric and polarimetric measurements by using inversion algorithms (e.g., Dubovik et al., 2021). Due to the ill-posed nature of the inverse problem, the inversion algorithms often use simplified forward scattering models and a priori assumptions to be able to retrieve aerosol properties. For instance, it is common to assume spherical shape and use Mie theory for the forward kernel of inversion algorithms (Holben et al., 1998;

Omar et al., 2009). While such an assumption is sufficient for spherical aerosols, studies have shown that light scattering by complex aerosols, such as biomass burning aerosols, is quite distinct from light scattering described by Mie theory (Espinosa et al., 2019; Manfred et al., 2018). Hence, aerosol property retrievals from remote sensing measurements are prone to uncertainties and biases. Independent in situ validation techniques are required to identify potential biases and uncertainties in the retrieved aerosol properties from remote sensing measurements (Mishchenko et al., 2007). Direct validation of remote

sensing measurements is a challenging and expensive task that often involves airborne measurements (Schafer et al., 2019). Alternatively, mimicking atmospheric remote sensing measurements with in situ instruments enables validation of remote sensing retrieval algorithms in laboratory environments (Schuster et al., 2019). This is a more cost-effective approach compared to the direct validation of atmospheric aerosol retrievals and allows for the testing of aerosol samples with well-defined properties.

Polar nephelometers are in situ instruments primarily designed for radiometric measurements. Following the Stokes formalism, polar nephelometers measure the $F_{11}(\theta)$ element of the aerosol scattering matrix $\mathbf{F}(\theta)$ over multiple scattering polar angles $\theta$ (Espinosa et al., 2017). The $F_{11}(\theta)$ element is also referred to as the phase function (PF) and describes the partial scattering coefficient of an aerosol as a function of $\theta$ for non-polarized incident light. A subset of polar nephelometers is also capable of performing polarimetric measurements, that is additionally providing the polarized phase function (PPF), - $F_{12}(\theta)/F_{11}(\theta)$, which

describes the relative degree and orientation of linear polarization of scattered light as a function of $\theta$ for non-polarized incident light (e.g., Dolgos and Martins, 2014).

Polar nephelometers have a long history (e.g., Waldram, 1945) and over the years several different instrument designs have been introduced. Broadly, polar nephelometer designs can be categorized into three groups (Barkey et al., 2012). Goniometer-type nephelometers conduct radiometric and polarimetric measurements using a detector mounted on a rotatable arm (Li et al.,

2018; Horvath et al., 2018; Waldram, 1945). These instruments can achieve high angular resolution at the expense of relatively



low time resolution. Multi-detector type nephelometers use sensors mounted at fixed scattering angles (Barkey et al., 2007; Dick et al., 2007; Nakagawa et al., 2016). These instruments can provide rapid radiometric and polarimetric measurements while the number of probed scattering angles remains limited. Laser imaging type nephelometers image light scattered by an ensemble of aerosol particles within a laser beam onto a charged couple device detector (CCD) such that scattering angle and

position on the image are unambiguously related. These instruments can provide radiometric and polarimetric measurements with high angular and time resolution. While more sophisticated polar nephelometers capable of measuring more components of $\mathbf{F}(\theta)$ exist (Hu et al., 2021), the majority of polar nephelometers used in atmospheric applications typically measure either PF only, or PF and PPF. Polarimetric data provided by these instruments make it possibly retrieve aerosol properties using inversion schemes similar (e.g. Espinosa et al., 2019, Schuster et al., 2019, Boiger et al., 2022).

In spite of several advantageous features, the use of laser imaging nephelometers has been quite limited. One of the earliest versions is the polarized imaging nephelometer (PI-Neph) that was introduced by Dolgos and Martins (2014). The PI-Neph provides both radiometric and polarimetric measurements at three wavelengths over an angle range of 3 to 177˚ with angular resolution of 1˚. This instrument has been deployed in airborne field campaigns. For example, Espinosa et al. (2017) applied the Generalized Retrieval of Aerosol and Surface Properties (GRASP), which is a well-established retrieval algorithm, on

measurements conducted by the PI-Neph and successfully retrieved size distribution, RI, and shape properties of aerosol particles.

Since the inception of the PI-Neph, several other laser imaging nephelometers have been introduced. For instance, Bian et al. (2017) and Manfred et al. (2018) used laser imaging nephelometers with only radiometric capabilities to measure the PF of ambient aerosol particles and biomass burning aerosols, respectively. More recently, Ahern et al. (2022) introduced a laser

imaging nephelometer capable of conducting simultaneous polarimetric and radiometric measurements at two distinct wavelengths which is suitable for deployment on aircrafts.

In this study, a new laser imaging type nephelometer, referred to as uNeph, was designed and constructed jointly by Micos Engineering GmbH (Dübendorf, Switzerland) and the Laboratory of Atmospheric Chemistry at Paul Scherrer Institute (Villigen PSI, Switzerland). Considerable down-sizing was a design goal for the uNeph to facilitate deployment and operation

of the instrument in different settings. This work presents the uNeph including instrument description and experimental setup (Section 2), basic data processing procedures (Section 3), signal calibration approaches and error assessment (Section 4) as well as validation and a first basic application (Section 5).

## 2 Instrument description and experimental setup

### 2.1 uNeph

The uNeph is a bench-top laser imaging polar nephelometer. It consists of an optical box with dimensions 44 x 61 x 18 cm and PC as well as a few small control boxes kept externally. A schematic of the key elements of the uNeph instrument is shown





in Fig. 1. A solid-state laser provides linearly polarized light at 532 nm (~200 mW). Exchangeable neutral density (ND) filters are utilized to reduce the laser intensity in order to adjust instrument sensitivity to different levels.

The next elements serve to rotate the angle of the linear polarization state following the approach detailed in Dolgos and
Martins (2014). It is an assembly consisting of a liquid crystal variable retarder (LCVR) and a Fresnel rhomb which acts as a quarter wavelength retarder. Subsequently, the beam is passed through multiple irises to adjust the cross-sectional area of the beam and to reduce the level of undesired background light (e.g., straylight) in the scattering chamber (not shown in Fig. 1).

The laser beam is directed through a scattering chamber twice using a rooftop reflector in between (a rooftop design that maintains the linear polarization nature of the laser beam). The input window of the scattering chamber is placed at angle and
the small reflected part, not blocked by the AR-coating, is used to measure the input laser power (forward beam) with a photodetector. The power of the backward beam exiting the chamber after both passages is also measured with a second photodiode (an ND filter is used for protection). Since the laser intensity attenuation is expected to be minimal within the range of aerosol sample concentration tested in this study we only used the forward beam laser power measurements to compensate the measurements for laser amplitude variations.

The scattering chamber, also shown in Fig. 1, serves as the measurement cell. It is a metal chamber with a length of ~30 cm and a volume of ~2.5 L. The laser beam enters and leaves the cell through sealed windows.  The chamber has a removable lid which enables access to the beam path, e.g., for performing calibration tasks. A pressure ($p$), temperature ($T$), and relative humidity ($RH$) sensor is mounted inside the scattering chamber. Sample inlet and outlet are installed at opposite ends of the scattering chamber to flush it continuously with gas or aerosol samples. A detection unit consisting of an optical objective and
a camera is used to collect the scattered light. Light scatters inside the forward and backward beams in forward (scattering angles, $\theta = 0\text{-}90\degree$) and backward ($\theta = 90\text{-}180\degree$) directions, respectively, and appears as two distinct stripes on the camera image (Fig. 2a). The camera is an Argon filled, actively cooled, monochrome, charged coupled device (CCD) with a resolution of 1392×1040 pixels (9×6.7 mm sensor size and 6.45×6.45 µm pixel size) and a 16 bit A/D converter (Trius Pro 825, Starlight Xpress Ltd.). A three-dimensional scheme of the laser path within the scattering chamber is demonstrated in Fig. S1. One
design element of the uNeph is the use of a wide field of view pinhole lens in the camera objective, which enables the instrument to be downsized. Based on the geometry (Fig. S1), the pinhole together with a given laser beam define a scattering plane. The forward and backward laser beams were aligned to be parallel, and the pinhole location was adjusted such that the two scattering planes have 45˚ (forward beam) and 135˚ (backward beam) orientations relative to the yz plane as depicted in Fig. S1b. They are chosen to be perpendicular to each other to achieve identical angle between scattering plane and orientation
of linear polarization state for both forward and backward beams. Two distinct linear polarization states, e.g., with orientation in parallel or perpendicular to the scattering planes, are sufficient to measure the phase function and polarized phase functions (see Sections 4.1). A polarimeter (PAX1000VIS/M, Thorlabs Inc.) was used to determine the two input settings for the LCVR that result in nominally fully parallel (∥) and perpendicular (⊥) linear polarizations. We use subscripts "1" and "2" to denote nominal parallel and perpendicular linear polarization operation set points, respectively. The raw data provided by the uNeph
consist of a digital image of the light scattered from the forward and backward beams at a given state of polarization. When



conducting measurements, a defined polarization state is first applied, followed by acquisition of an image (or multiple images) at a specified exposure time, $t_{expo}$. Auxiliary sensor readings ($T$, $p$, $RH$, and photo detectors) are simultaneously being logged. Figure 2a shows an example of a raw light scattering image for a particle-free air sample taken at polarization state 2, i.e., nominally perpendicular linear polarization. Further data processing will be explained in Section 3 onwards.

## 2.2 Experimental setup for uNeph calibration and validation

Some uNeph calibration measurements rely on probing pure, particle-free gases following the experimental setup shown in Fig. 3a. For this purpose, particle-free air, $CO_2$, Ar or He was flushed through the uNeph with a flow rate of ~ 5 Lmin$^{-1}$. Further calibration and validation measurements were done using aerosol samples with well-defined properties using the experimental setups shown in Fig. 3b or Fig. 3c to generate quasi-monodisperse or broad unimodal aerosol samples, respectively. Initial aerosol generation steps were identical for these two types of experiments. The aerosol samples used in this study were spherical polystyrene latex size standards (PSL; see Table S1 for specifications) and Di-Ethyl-Hexyl-Sebacate (DEHS). DEHS is a non-absorbing oil-like liquid, hence also resulting in spherical aerosol particles.

PSL suspensions diluted with Milli-Q water were nebulized using a commercial atomizer aerosol generator (ATM 226, Topas GmbH, Dresden, Germany). Liquid DEHS in pure form was aerosolized using a Collison-type nebulizer (CH Technologies; Westwood, NJ, USA). After particle generation, aerosol samples were passed through a Kr-85 neutralizer to mitigate aerosol electrostatic losses in the sampling line. Subsequently, to remove water content from the nebulized aerosol particles, the sample flow was passed through a silica gel diffusion drier. A dilution stage was placed after the drier to adjust aerosol concentration to suitable levels, particularly during the tests with DEHS for which initial particle number concentrations were typically too high. Next, a quasi mono-disperse size cut was extracted from the sample flow by directing it through an aerosol aerodynamic classifier (AAC, Cambustion, Ltd., Cambridge, UK; Tavakoli & Olfert, 2013). The major advantage of the AAC is that the size selection only depends on particle aerodynamic diameter while being independent of charge, i.e., there is no interference from larger particles carrying multiple charges. For PSL aerosol experiments, the AAC was operated at a resolution parameter set point of 10 and the nominal diameter setpoint of the AAC was adjusted to maximise particle number concentration downstream, i.e., at the peak of the PSL size mode. This approach ensures that unwanted small residual particles from solutes in the suspension and possible agglomerated multiplets are removed without causing a shift in modal size of the selected PSL particles. The AAC set point diameters which maximized PSL transmission agreed within 1 % to the calculated aerodynamic diameters of the PSL size standards (Table S1), which validates accuracy of size selection by the AAC. For the DEHS aerosol experiments, the AAC was operated at a resolution parameter set point of 20 to provide a quasi mono-disperse DEHS aerosol of known size. The AAC aerodynamic diameter set points and corresponding volume equivalent diameters are included in Table S2. In all the tests the sample flow through the AAC was ~ 1 L min$^{-1}$. So far, particle generation and selection were identical for all aerosol experiments. For the mono-disperse test experiments (Fig. 3b) the size-selected aerosol sample was combined with a stream of particle-free compressed air which was maintained at ~4 L min$^{-1}$, before being sent to the uNeph



and a condensation particle counter (CPC, TSI Inc., Shoreview, MN, USA, Model 3776) operated in parallel. This provides phase function measurements for an aerosol of known refractive index (RI), size and number concentration.

The experimental setup was slightly modified to generate and probe a broad unimodal aerosol with larger but still moderate width of the size distribution (Fig. 3c). A holding container (with volume of ~100 L) was placed after the aerosol classification stage. The AAC was stepped through 6 different set points ranging from 310 nm to 450 nm aerodynamic diameters while filling the container in flow-through mode over a time period of ~10 min. Afterwards, the outlet valves were closed for ~18 hours to allow the size distribution to become more homogeneous through mixing and coagulation. After the coagulation

process, the sample was slowly pushed through the outlet of the holding container by applying a particle free air flow of 0.5 L min$^{-1}$ at the inlet. The extracted aerosol sample was diluted with 4 L min$^{-1}$ particle-free air and then distributed to the aerosol instruments. In addition to the uNeph and the CPC, a scanning mobility particle sizer (SMPS) was employed to measure the aerosol number size distribution. The SMPS was a combination of a differential mobility analyser (DMA, TSI Inc., Shoreview, MN, USA, Model 3082) and a CPC (TSI Inc., Shoreview, MN, USA, Model 3775).

## 175    3 uNeph data processing

Deriving calibrated phase function and polarized phase function data from the uNeph raw digital images requires many intermediate data processing steps. The two main stages are image data reduction and angular signal processing as delineated in flowcharts (Fig. S2) and detailed in the following.

### 3.1 Image data reduction

#### 180    3.1.1 Dark signal corrections

Digital images acquired by the CCD contain some signal that is entirely unrelated to actual illumination of the CCD, hereafter referred to as *dark signal*. It can be characterized by acquiring images without illuminating the CCD (Manfred et al., 2018), i.e., with the uNeph laser turned off. The CCD also has a few *hot pixels* that possess abnormally large dark current (e.g., Fig. S3). Image processing, which follows the flowchart in Fig. S2a, starts with hot pixel identification and removal (Fig. S2a),

as detailed in Section A1. Subtraction of dark signal contribution as detailed in Section A2 comes next. The dark signal itself has two systematic contributions, a positive bias (constant) and dark current (proportional to exposure time), as well as superimposed random noise. Accordingly, two constants are sufficient to characterize the systematic components of the dark signal as a function of the exposure time (Eq. A2). Figure S4 demonstrates that this correction approach works well to subtract dark signal contribution with small residuals.

#### 190    3.1.2 Scattering angle calibration calibration

The relationship between scattering angle and image pixels is determined through the scattering angle calibration, which is described in more detail in Section A3. Briefly, this is achieved by relating an axial position inside the laser beam with both





the scattering angle and the corresponding image pixel coordinates. For this purpose, a pinhead mounted on a 3D translation stage is placed at several positions inside the laser beam (Fig. S5a). An image of the diffused laser light is recorded along with

the corresponding coordinates of the pinhead position. Figure S5b illustrates how the laser beam axis, the pinhead position (*S*), and the center of the pinhole objective (*P*) define the scattering angle ($\theta$). The center of the pinhole objective *P* is not known exactly, which introduces uncertainty into $\theta$ as explored and discussed in Section 4.3. The center of the bright spot on the image provides the corresponding pixel (Fig. S6a). This calibration step ultimately provides a list of pixel coordinates along the centerline of the forward and backward beams together with corresponding scattering angles. Such calibration points are

shown in Fig. 2b as yellow dots along with a second-order polynomial fit curve through them (red curve).

### 3.1.3 Angular signal extraction

The laser beam stripes on the image are wider than just one pixel (e.g., Fig. 2b). Limiting further data analysis steps to the centerline pixels would impose unnecessarily high statistical noise in the results (Ahern et al., 2022). Therefore, the next steps, following the flowchart shown in Fig. S2a, aim at integrating the raw image signal along the beam cross-section for each angle.

The image is first transformed to bring the laser beam stripes on a straight line in parallel to the new abscissa representing scattering angle. The top (backward beam) and bottom (forward beam) halves of Fig. 2b are separately transformed and stitched together with a common abscissa as shown in Fig. 2c and described in more detail in Section A4. The blue lines in these two panels illustrate the beam cross-section at $\theta$=40° before and after transformation as an example.

The transformed image signal is then integrated along the beam cross-section for each angle, illustrated in Fig. 2d for $\theta$=40°

as an example. Integration boundaries were chosen to maximize the ratio of actual light scattering signal to interfering signal contributions such as stray light or dark current residuals. Specifically, we chose the integration limits, indicated by red lines in Figs 2c and 2d, at the pixels where the signal drops to ~10% of the peak signal at the beam center (see Section A4 for details). Integration at each angle provides, at the end of the flowchart shown in Fig. S2a, an angle-dependent scalar value, which represents the raw angular signal, $\Xi(\theta)$.

### 3.2 Angular signal processing

#### 3.2.1 Selection of valid signals

The initial data processing steps described in Section 3.1 serve to provide a signal that is, ideally, strictly proportional to image exposure time and without offset. Two signals $\Xi_1(\theta)$ and $\Xi_2(\theta)$ obtained when probing a stable homogeneous scattering medium with two different exposure times $t_{expo,1}$ and $t_{expo,2}$, respectively, are expected to fulfil:

$\Xi_2(\theta) = \Xi_1(\theta) \times t_{expo2}/t_{expo1}.$ (1)

In practice, this proportionality relation deteriorates at too small signals, when residuals of dark signal become relevant, or at too high signals, when saturation occurs. Therefore, the data processing contains a step to discard invalid signals outside the strictly proportional range (see flowchart in Fig. S2b). To estimate the valid signal range across which proportionality holds,





we measured particle-free air for 12 different $t_{\text{expo}}$ covering the range from 0.1 s to 464 s. We calculated the expected $t_{\text{expo}}$-

dependence of the angular signal, denoted as $\Xi_{\text{exp}}(\theta, t_{\text{expo}})$, with assuming that Eq. 1 holds and with choosing the largest valid

signal below onset of saturation as a reference point. This allows for testing proportionality to $t_{\text{expo}}$ by comparing the actual

measured signal, denoted as $\Xi_{\text{meas}}(\theta, t_{\text{expo}})$, against $\Xi_{\text{exp}}(\theta, t_{\text{expo}})$ as shown in Fig. 4a for $\theta=50°$. This demonstrates that

proportionality is fulfilled in principle. Only a more precise assessment done in Fig. 4b, which presents the relative deviation

$\frac{\Xi_{\text{meas}}-\Xi_{\text{exp}}}{\Xi_{\text{exp}}}$, reveals the limits of proportionality. Systematic low bias due to saturation occurs for $\Xi > {\sim}7{\cdot}10^5$. Bias also increases

for very low signals, e.g., relative deviation exceeds 5% for $\Xi < {\sim}2{\cdot}10^3$ in this example. Detailed results for a wide range of

angles are shown in Fig. S8 and Fig. S9 for proportionality tests with cooled and uncooled CCD, respectively. Generally,

proportionality was fulfilled down to lower values of $\Xi$ when cooling the CCD due to smaller residuals of dark signal

subtraction (Fig. S10).

Figure S11 shows repeated measurements of particle-free air samples taken over a wide range of exposure times. The two

panels with longest exposure times (bottom right) demonstrate how the upper limit (saturation) of the A/D converter leads to

capping of the signal (red line) in the centre part of the beam cross-section, which causes a systematic low bias in the integrated

value (blue marker, right axes). At all other exposure times ($t_{\text{expo}} \leq 100$ s), the integrated signal has a high precision (blue

markers and error bars), though random noise does become important at $t_{\text{expo}} < {\sim}1$ s (for particle-free air at the considered

scattering angle).

Above results from testing proportionality of signals to $t_{\text{expo}}$ were used to define lower and upper limits of raw $\Xi$ outside which

signals are discarded. We only retain signals for which the relative deviation remains below ~6% (lower limit) and for which

no saturated pixel occurs (upper limit). Applying these strict limits approximately leaves a dynamic range of ~2-3 or ~1-2

orders in magnitude for cooled or uncooled CCD, respectively (Fig. S10).

Unfortunately, three orders in magnitude of dynamic range may not be sufficient to operate the uNeph with a fixed $t_{\text{expo}}$ as the

signal strength can vary over an even wider range depending on scattering angle, polarization setting and sample properties.

For example, the signal at around 90° drops to very low values compared to forward/backward scattering when applying

parallel polarization to a Rayleigh scatterer, or forward scattering can exceed backward scattering by orders in magnitude for

large particles. When measuring aerosol samples, additional and considerable variation in the signal $\Xi$ can occur due to

statistical fluctuations of the number of particles present inside the laser. Figure S12 presents signals for measurements with a

high product of DEHS particle number concentration and exposure time, which results in homogeneous signals. Here we mean

homogeneous in the sense that light scattered out of the laser beam appears as smooth stripes (Fig. S12a), and random noise

of repeated measurements remains on a low level (represented by the spread of the grey curves in Fig. S12b). Consequently,

the histogram of repeatedly measured $\Xi$ values for the aerosol sample (blue bars in Fig. S12c) has a narrow width. This implies

that the mean number of particles present in the sensitive volume of the laser for this angle during image exposure is similar

for all repeats with little statistical fluctuations. The mean value of $\Xi$ is much higher for the aerosol sample compared to the



particle-free air sample shown for comparison (red bars), because the aerosol scattering coefficient clearly exceeds the air scattering coefficient at this particle size and concentration.

By contrast, Fig. S13 shows identical measurements as in Fig. S12, but for DEHS particle number concentration being a factor of 1000 lower. In this example, the product of particle number concentration times exposure time is small enough to cause inhomogeneous signals. The stripes in Fig. S13a are not anymore smooth and instead bright spots along the laser beam become discernible. These are caused from single particles crossing the beam during image exposure. Accordingly, the signal along the beam cross-section also has high fluctuations between repeats (spread in the grey lines in Fig. S13b). In some repeats, the signal remains at the level of the air background (red line) across the entire beam cross-section or portions of it. The histogram of $\Xi$ values shows that the aerosol sample signal (blue bars in Fig. S13c) is identical to the air background (red bars) for a large fraction of the images because no particle crosses the laser beam at this angle during the exposure. A subset of the $\Xi$ values is considerably larger, corresponding to cases when a particle is present during the exposure. For this example, the minimal and maximal signal of single images differ by a factor of ~3 due to random fluctuations caused by limited particle statistics. This problem can be mitigated by averaging sufficiently large number of images with identical exposure time (red dashed line in Fig. S13b).

Given this issue of sample homogeneity, probing the full phase function with small random noise may make it necessary to measure with different exposure times and to include repeats at each of them. A condition for obtaining an unbiased average is that all repeats taken at a given $t_{\mathrm{expo}}$ fall within the proper signal range for a given angle. If some repeats were falling outside the proper signal range for a given angle, then the average would likely be biased. No bias is expected if either all images are retained or if all invalid images are discarded at that angle. Therefore, all measurements taken at a $t_{\mathrm{expo}}$ with some invalid signals are to be excluded from further data analysis. Discarding data outside proper signal range, is implemented as the first data processing step starting from the integrated signals $\Xi$ (see flowchart in Fig. S2b).

In the presence of large particles, the data filtering step may disqualify all $t_{\mathrm{expo}}$. This is due to the fact that the difference in signal with or without a particle present in the laser beam becomes increasingly large with increasing particle size. Above a certain size this difference can exceed the proportional range of the instrument, such that it is not possible to find an exposure time for which all signals are unbiased. At longer $t_{\mathrm{expo}}$, images with particles present suffer from saturation. At shorter $t_{\mathrm{expo}}$, images without particles present fall below the proper signal range.

One approach to mitigate the unresolvable $t_{\mathrm{expo}}$ trade-off is reducing instrument sensitivity by, e.g., reducing laser power with a stronger ND filter. This would allow for longer $t_{\mathrm{expo}}$ without exceeding the saturation limit. Longer $t_{\mathrm{expo}}$ reduces signal fluctuations related to particle statistics, such that also the smallest signals remain above the lower signal limit.

### 3.2.2 Normalization of the signal by exposure time and laser power

The signal $\Xi$ is proportional to $t_{\mathrm{expo}}$, a data acquisition setting, and to laser power. Hence, the next data processing step is normalization of $\Xi$ by $t_{\mathrm{expo}}$ and the laser power signal (flowchart in Fig. S2b), in order to account for variations in these parameters. The forward beam photodetector provides a signal proportional to the laser power by probing the part reflected at



the chamber window (Fig. 1). The normalized signal $\varXi$ is referred to as the compensated signal and is denoted as $\xi(\theta)$. This

normalization step allows for averaging $\xi(\theta)$ acquired at different laser power and with different $t_{\mathrm{expo}}$.

### 3.2.3 Subtracting stray light interference

The compensated signal $\xi(\theta)$ contains a contribution from stray light background (e.g., light scattered from the walls of the sampling volume) which interferes with the light scattered by the sample. The next data processing step aims at subtracting this interference (flowchart in Fig. S2b). In this study we determined the stray light signal contribution by sampling helium

(He) gas with the uNeph. The scattering coefficient of helium is more than sixty times smaller than that of air, such that $\xi(\theta)$ measured for a helium sample typically is dominated by stray light contribution. Therefore, it is commonly used to quantify stray light interference (Ahern et al., 2022; Manfred et al., 2018). We denote the normalized stray light signal as $\xi_{\mathrm{SL}}(\theta)$ and subtract it from $\xi(\theta)$ to obtain the sample signal, $\xi_{\mathrm{meas}}(\theta)$. Thus, $\xi_{\mathrm{meas}}(\theta)$ for an aerosol sample only contains contributions from light scattered by the carrier gas and the particles (plus residuals from dark signal and stray light corrections). Stray light does

not depend on temperature or pressure. Therefore, correction of this interference is kept separate from air background subtraction (described in Section 3.2.5).

### 3.2.4 Signal averaging to mitigate random signal fluctuations

For the reasons explained in Section 3.2.1, we typically acquired repeat measurements at different $t_{\mathrm{expo}}$ values. The data processing steps done so far (Fig. S2b) provide $\xi_{\mathrm{meas}}(\theta)$, which can be averaged over repeated measurements without further

corrections. We apply a weighted averaging in which individual $\xi_{\mathrm{meas}}(\theta, t_{\mathrm{expo}})$ are weighted by their corresponding $t_{\mathrm{expo}}$ in order to obtain $\xi_{\mathrm{mean}}(\theta)$. The weighting is introduced as shorter measurements of the same sample are expected to have poorer signal to noise ratio. The results for the air samples presented in Fig. S11 justify this approach. First, the random noise in single measurements is negligible for exposure times of ~21.5 s or longer, whereas it is considerable for exposure times of ~2.15 s or shorter. Second, the mean results from repeated short measurements (red lines & blue markers) are consistent with the

results at long exposure times.

### 3.2.5 Air background subtraction

For a particle-free gas sample, the preparatory data processing steps (flowchart in Fig. S2b) are complete after stray light subtraction and optional averaging, i.e., $\xi_{\mathrm{gas}}(\theta) \equiv \xi_{\mathrm{meas}}(\theta)$, which only leaves application of the calibration constants as a final step to follow (Section 4). In contrast for an aerosol sample, the signal $\xi_{\mathrm{meas}}(\theta)$ is proportional to the sum of the light scattered

by particles as well as the light scattered by the carrier gas (in this case air) present in the laser beam. Thus, subtraction of the air background contribution, $\xi_{\mathrm{BG}}(\theta)$, is an additional step that is required to derive the signal, $\xi_{\mathrm{aerosol}}(\theta)$, that is proportional to the light scattered by the aerosol particles only (flowchart in Fig. S2b). We apply the approach of regular filtered air measurements in order to obtain a reference value for the air background signal $\xi_{\mathrm{BG}}(\theta)$. Data processing for this air background measurement follows the standard *gas branch* of the flowchart in Fig. S2b to obtain $\xi_{\mathrm{air}}(\theta)$. An aerosol measurement is taken



at a certain temperature ($T$) and pressure ($p$), whereas the air background is measured at potentially different conditions ($T_{ref}$, $p_{ref}$). Given that the scattering coefficient of air depends on temperature and pressure, the following correction is applied to $\xi_{air}$;

$$\xi_{BG} = \xi_{air}\frac{T_{ref}}{T}\frac{p}{p_{ref}}. \tag{2}$$

The magnitude of air signal variability in the time window between two air background measurements determines the residual

error in the aerosol signal due to imperfect background subtraction. Therefore, we conducted continuous air background measurements over an extended period of time in order to estimate this variability. The variations of $\xi_{BG}$ relative to an arbitrarily chosen reference value $\xi_{ref}$ are presented in Fig. S14 and Fig. S15 for the polarization set points 1 and 2, respectively. In these examples with a duration of ~3 days, the systematic drift dominates over random noise, while remaining within a few percent. Figure S16 shows eight air background measurements distributed over 14 days. These results suggest an instrument stability

of around ±3% over this duration. This means that residuals from imperfect air background subtraction contribute to random noise in $\xi_{aerosol}(\theta)$ on the level of ~3% of the air background signal.

## 4 Instrument calibration and error model

The uNeph data processing steps described in Section 3 provide a signal that is directly related to the light scattering phase function of either gas or aerosol samples. The last remaining step is to determine and apply a set of calibration constants to

derive phase functions in absolute units.

### 4.1 Calibration equations

The measurement and calibration approach applied in the uNeph to determine the absolute phase function ($F_{11}(\theta)$; [$Mm^{-1}sr^{-1}$]) and the polarized phase function ($-F_{12}(\theta)/F_{11}(\theta)$; [-]) builds up on the work by Dolgos and Martins (2014), i.e., on taking measurements at two well-defined laser polarization states (i.e., $\xi_1(\theta)$ and $\xi_2(\theta)$). Using the Stokes formalism, the following

pair of equations relates the measurements ($\xi_1(\theta)$, $\xi_2(\theta)$) to the scattering matrix elements ($F_{11}(\theta)$, $F_{12}(\theta)$) for a defined sample (either a gas or an aerosol) consisting of an ensemble of randomly oriented scatterers:

$$F_1(\theta) := G_1(\theta)\xi_1(\theta) = F_{11}(\theta) + q_1 F_{12}(\theta) , \tag{3a}$$
$$F_2(\theta) := G_2(\theta)\xi_2(\theta) = F_{11}(\theta) + q_2 F_{12}(\theta) . \tag{3b}$$


Here, $G_1(\theta)$ and $G_2(\theta)$ are instrument gain calibration factors for the two polarization set points, whereas $q_1$ and $q_2$ represent true fractions of linear polarization aligned with the nominal orientation of the linear polarization states. For perfect polarization control, $q_1$ and $q_2$ assume the values +1 and -1, representing 100% linear polarization parallel and perpendicular to the scattering plane, respectively. However, polarization control and geometry are not perfect such that $q_1$ and $q_2$ are expected



to be smaller than +1 and larger than -1, respectively. In Eq. 3 we also define $F_i(\theta)$, which is the actual angular distribution of total scattered light in unit of Mm⁻¹sr⁻¹ for polarization set point $i$. To distinguish between the perfect and actual polarization states, we also define the terms $F_{\text{para}}$ and $F_{\text{perp}}$, to refer to hypothetically perfect measurements with $q_1 = 1$ and $q_2 = -1$, respectively.

**4.2 Radiometric calibration using gas samples**

The first step in the calibration process is the evaluation of the gain calibration factors. In our analysis, we used particle free air and $CO_2$ as calibration gases. For the calibration gases, $F_{11}(\theta)$ and $F_{12}(\theta)$ are taken from the literature (Dolgos and Martins, 2014). Initially, we assumed perfect polarization set points, that is $q_1 = 1$ and $q_2 = -1$. Then we obtained $G_i(\theta)$ using Eq. 3 and $\xi_i(\theta)$ measured for the calibration gases. Specifically, we used the difference $\xi_{i,CO2}(\theta) - \xi_{i,air}(\theta)$ to avoid interference from residual dark signal or stray light contributions to $\xi_i(\theta)$. The magenta lines in Fig. S17 show the resulting angle-dependent gain

calibration factors. Additionally, gain factors derived from single gas calibration measurements, i.e., with either using $\xi_{i,CO2}(\theta)$ (red lines) or using $\xi_{i,air}(\theta)$ (blue lines) are also shown. The fact that all three curves are very similar demonstrate that the residual signal offset is very small, and that our gain calibration has a high precision.

In a next calibration step, we measured argon (Ar) gas to examine the validity of the assumed $q$ values. Ar is a monatomic gas for which $F_{\text{para}}(\theta)$ approaches zero as $\theta$ approaches 90˚ (Fig. S18). This makes it ideal to reveal errors in angle calibration or

$q_i$. Indeed, the comparison of uNeph measurements and theoretical curves in Fig. S18 (and its variant Fig. S19 zoomed in at $\theta \approx 90°$) suggests some bias. Applying the gain calibration, obtained with the assumption $q_1 = 1$, results in systematically greater uNeph measurement compared to the theoretical values in the angle range of ~ 75˚ to 105˚. This suggests that the uNeph polarization control is not as perfect as assumed. Figures S18 and S19 additionally contain curves obtained with assuming a range of different $q_i$ values. The measurements obtained by assuming $q_1 = 0.92$ (in the forward angular direction) and $q_1 = 0.95$

(in the backward angular direction) closely match the theoretical results. Therefore, we used these $q_1$ values for the subsequent data analyses. Furthermore, we also assumed that $|q_1| = |q_2|$. This is a simplified approach to calibrate the actual polarization states. Using Ar validation data alone is not sufficient to disambiguate the bias from $q$ calibration and the bias from angle calibration. Therefore, we used a PSL aerosol with well-constrained properties to further optimize the uNeph calibration.

**4.3 Refining calibration using PSL size standard**

The radiometric calibration and the scattering angle calibration are inter-connected through Eq. 3. The $q$ values were identified as a main source of uncertainty in radiometric calibration (Section 4.2). The exact position of the camera's pinhole is the main source of uncertainty in angle calibration (Section 3.1.2). In order to refine these calibration parameters, we use a PSL aerosol with well-constrained properties as a further calibration reference. Using PSL standards as a calibration step was previously implemented by Ahern et al (2022).

In this study, the monodisperse PSL aerosol was generated as described in Section 2.2. The expected absolute phase functions were calculated using Mie theory, with RI taken from the literature (Kasarova et al., 2007; Ma et al., 2003), and assuming a





lognormal size distribution. We use the MiePython package (https://github.com/scottprahl/miepython) to carry out the Mie calculations. The lognormal parameters were taken from the certified diameter (600 nm), reported coefficient of variation (1.7%) and number concentration measured by the CPC.

For a given set of $q$ values and pinhole location $P$, it is possible to process the uNeph data all the way to absolute phase functions. By varying the value of $P$ for fixed $q$ values it is possible to optimize the angle calibration by choosing the value which results in the smallest least-squares sum of residuals between expected and measured phase functions. Changing angle calibration, deteriorates agreement for the Ar data, which were used to optimize $q$ values. Therefore, $q$ values and pinhole location were alternately optimized in a few iterations.

Figure S20 illustrates the effect of optimizing the coordinates of the pinhole center. The dashed red-lines are the uNeph data processed with the final optimized $q$ values along with the initial coordinates of the pinhole center taken from the angle calibration process (Section 3.1.2). The solid red lines are the uNeph data processed with the final optimized $q$ values along with final optimized coordinates of the pinhole center. Comparison of these phase functions against the expected phase functions illustrates the considerably improved agreement after optimization.

The fact that good agreement is achieved at the end of this optimization process for one specific calibration aerosol, does not necessarily imply that the calibration constants are physically meaningful. Therefore, the uNeph measurement still needs to be validated using aerosol samples with well-constrained properties. Such validation results are presented in Section 5.

It can be seen in Fig. S20 that the final calibrated phase function data have a gap in the angle range between ~85° and ~95°. The angle of the camera's central axis relative to the laser beams was not optimally chosen in the uNeph prototype such that

the portions of the forward and backward beams corresponding to this angle range fall outside the camera's field of view. This type of angular truncation (side angle truncation) is unfortunate, however, it does not substantially affect the retrieval of aerosol parameters from uNeph measurements, as shown in Moallemi et al. (2022).

**4.4 Measurement error assessment**

To better understand and quantify the errors in uNeph measured phase functions, we developed an instrument error model.

This model contains the major uNeph error sources as well as uncertainty values for the parameters that govern each type of error. As demonstrated in Sections 4.2 and 4.3, uncertainties in the exact laser polarization state and angle calibration are important sources of uNeph measurement error. Air background subtraction and measurement precision also contribute to the measurement error. We account for these four independent sources of error and consider their contributions to the total phase function measurement error ($\sigma_{\text{tot}}$) to be independent of each other. Hence, we combine them following standard error

propagation for independent errors:

$$\sigma_{\text{tot},l}(\theta) = \sqrt{\sigma_{\theta,l}^2(\theta) + \sigma_{BG,l}^2(\theta) + \sigma_{q,l}^2(\theta) + \sigma_{\varepsilon,l}^2(\theta)} \, , \tag{4}$$





where $\sigma_{\theta,l}$, $\sigma_{BG,l}$, $\sigma_{q,l}$, and $\sigma_{\varepsilon,l}^2$ represent individual contributions to phase function error arising from uncertainties in angle calibration, background subtraction, polarization state calibration, and signal precision, respectively. The subscript $l$ is a placeholder denoting errors of $F_1$, $F_2$, $F_{11}$ or $-F_{12}/F_{11}$. A detailed description on the evaluation of different error components is provided in the section A5 of the appendix.

The total measurement error, $\sigma_{tot,l}$, obtained using the error model based on Eq. 4 and its constituting components, are shown in Fig. 5 for the 600 nm PSL aerosol test case. The black lines in panels e-f of Fig. 5 demonstrate that the total measurement error strongly depends on angle due to variable contributions from individual error components. In this example, estimated total error remains below 10% in $F_{11}$ for most angles, and below 0.1 (absolute) in $-F_{12}/F_{11}$ for all angles. The complexity of the uNeph measurement errors make it necessary to use an error model for precise error estimates as a function of phase function shape, aerosol concentration and scattering angle. The 600 nm PSL aerosol test case, which was used to illustrate how different components contribute to measurement error, is not a rigid test for the error model given that these measurement data were also used to refine the angle calibration (Section 4.3). Therefore, further validation of estimated error magnitudes is presented for the uNeph validation experiments discussed in Section 5.2.

## 5 Instrument validation and example application

### 5.1 Validation of phase function absolute values

The calibration approach for the uNeph described in Section 4.2 is designed to provide phase matrix elements $F_{11}(\theta)$ and $F_{12}(\theta)$ in absolute units ([Mm$^{-1}$·sr$^{-1}$]), as opposed to just providing normalized phase matrix elements $P_{11}(\theta)$ and $P_{12}(\theta)$ (unit: [sr$^{-1}$]). Here, we assess the level of accuracy of the measured absolute values, which depends on the accuracy of the gain calibration factors $G_i$ (Eq. 3), as well as the precision in compensated signal $\xi$. To do so, we used mono-disperse PSL size standards with diameters of 240 nm and 600 nm and the experimental setup shown in Fig. 3b. Validation was done for $F_1(\theta)$ and $F_2(\theta)$ measured by the uNeph. The size distribution parameters (modal size and width) and RI values are fixed and known for PSL aerosols, thus making measured $F_i(\theta)$ strictly proportional to particle number concentration at any angle. Therefore, it is possible, using Mie theory, to infer particle number concentration directly from $F_i(\theta_j)$ measured at a single angle $\theta_j$. This was done for all angles with valid uNeph measurements, for the two polarization set points and for the two PSL sizes. Statistics of particle number concentration values determined with this approach are provided in Table 1 (4$^{th}$ and 5$^{th}$ column). The coefficients of variation (CV) of the uNeph-derived particle number concentrations over $\theta$ were as low as 4%, 12%, 5% and 2% for the two PSL sizes and polarization set points. These results demonstrate that precision of the gain calibration for single angles is very high, given that errors in PSL size distribution properties and other random noise can also influence the coefficients of variation. The relative bias of mean uNeph-derived particle number concentrations compared against independent measurements by a CPC is listed in the last column of Table 1. Agreement is excellent with bias ranging from --



4.6% to +3%. This is actually much better than the specified CPC uncertainty of ~10%. Altogether we conclude that the gain calibration is very precise and that there is no evidence of systematic bias that goes beyond CPC uncertainty.

### 5.2 Validation of phase functions using quasi-monodisperse aerosol

The uNeph data presented in Fig. 5 do not validate its performance for measuring any type of phase function, because this experiment was also used to refine the angle calibration. Therefore, uNeph performance was further validated by probing quasi-monodisperse spherical aerosol particles with known complex RI and diameters of 200 nm, 400 nm, 600 nm and 800 nm. These validation aerosols were generated by extracting a narrow size cut from a broad unimodal DEHS aerosol by means of an AAC (see Fig. 3b for experimental setup). The sampling period was ~ 60 min for each size and included the recording of

multiple repeats of data at different exposure times (0.1 s ≤ $t_{expo}$ ≤ 100 s) to ensure that valid signals were collected for all angles and polarization set points (Section 3.2.1). The aerosol number concentration was quite stable and small drifts during the experiments were accounted for in the data analysis.

DEHS is a liquid, which therefore results in spherical particles when aerosolized. This makes it possible to use Mie theory to calculate expected light scattering phase functions. Nevertheless, the true phase function is not exactly known, due to

uncertainties in the aerosol parameters that are required for input in the Mie calculation. A complex RI of 1.455 +0$i$ at 532 nm was used based on Pettersson et al (2004). The AAC classified particles were assumed to have log-normal size distributions with best estimates for modal diameter inferred from AAC aerodynamic diameter setpoints (see Table S3) and for number concentration taken from the CPC measurement. To account for uncertainties in the properties of the AAC-extracted aerosol, a Monte Carlo simulation was performed to calculate a range of expected phase functions by varying the size distribution

parameters. The complex RI was held fixed and normally distributed errors were assumed for modal diameter and number concentration with coefficients of variation equal to 3% and 10%, respectively. The width of the lognormal size distribution, expressed as geometric standard deviation (GSD), is not exactly known. Therefore, possible GSD values were assumed to be evenly distributed between the limits 1.04 and 1.08. $F_{11}$ and $-F_{12}/F_{11}$ were calculated for 1000 sets of randomly drawn size distribution parameters as described above, and the interquartile range of all resulting phase function values at a given angle

is assumed to represent uncertainty of the true phase function as constrained by AAC and CPC.

The uNeph validation results are presented in Fig. 6. The uncertainty range of expected $F_{11}$ and $-F_{12}/F_{11}$ is shown as grey shadings, and the measured phase functions are shown as red lines with error bars calculated with the error model presented in Section 4.4. The $F_{11}$ measurements fall well within the uncertainty range of expected phase functions over all available angles and for all four sizes (top panels). More precisely, most data points fall into the uncertainty range of the predictions

even without error allowance on the uNeph measurement. There is hardly any disagreement between expected and measured $F_{11}$ that exceeds the measurement errors. The findings are quite equivalent for the $-F_{12}/F_{11}$ function: the measurements fall well within the uncertainty range of expected phase functions (bottom panels in Fig. 6), for most part even without allowance





for measurement error. Disagreement that exceeds the estimated measurement errors only occurs at a few angles for the 800 nm particles. The uncertainties in the true phase functions, i.e. the widths of the grey shadings, are quite considerable despite using

well-defined reference aerosols. This impedes stringent test of the error model, i.e. reliable identification of potentially underestimated measurement errors.

An alternative approach to validate the phase function measurements is to use them to retrieve the properties of the test aerosol size distributions, which are assumed to be of lognormal shape and described by the vector $\boldsymbol{v}_{PSD}$ defined in Eq. 5a with the elements geometric mean diameter ($d_{\mathrm{m}}$), geometric standard deviation ($GSD$), and total particle number concentration ($N_{\mathrm{tot}}$).

$$\boldsymbol{v}_{psd} = \begin{bmatrix} d_{\mathrm{m}} \\ GSD \\ N_{\mathrm{tot}} \end{bmatrix} \tag{5a}$$

For this purpose, a simple retrieval scheme was applied to retrieve $\boldsymbol{v}_{PSD}$, optimized such that the corresponding phase functions calculated with Mie theory achieve best fit to the phase function measurement data, specifically F1 and F2, according to the least square minimization given in Eq. 5b.

$$\boldsymbol{v}_{psd,\ fit} = \min_{\boldsymbol{v}_{psd}}\left(\frac{\sum_{\theta_1}^{\theta_N}\left(\ln(F_{1,meas}(\theta_i)) - \ln(F_{1,Mie}(\theta_i,\boldsymbol{v}_{psd}))\right)^2 + \sum_{\theta_1}^{\theta_N}\left(\ln(F_{2,meas}(\theta_i)) - \ln(F_{2,Mie}(\theta_i,\boldsymbol{v}_{psd}))\right)^2}{N}\right) \tag{5b}$$


The complex refractive index of DEHS at 532 nm was again taken as 1.455 +0$i$ (Pettersson et al., 2004). The blue lines in Fig. 6 correspond to $F_{11}$ and -$F_{12}/F_{11}$ calculated with the best fit $\boldsymbol{v}_{PSD}$. These curves match the measurement data within measurement error except for very few data points. This shows that the shape of the measured phase functions is physically meaningful. The corresponding retrieved size distribution parameters are included in Table S3 along with independent data. The retrieved GSD

values ranged from 1.035 to 1.065, which is very narrow but within a plausible range for the given AAC resolution parameter settings. The retrieved number concentrations agree with the CPC data to within -2% to +6%, which falls within the uncertainty range of the CPC. The retrieved diameters agree with the AAC data to within -0.5% to -3.8%, which is very good agreement, though located at the edge of expected AAC uncertainty. The fitted phase functions corresponding to the best fit $\boldsymbol{v}_{PSD}$ (blue lines in Fig. 6) also fall within the uncertainty range of the true phase functions as indicated by the grey shadings. The course

of the blue lines within the grey shading areas is not random. They closely follow one edge of the shading for the three AAC set points shown in Panels c) to h). Figure S24 is identical to Fig. 6 with additional examples of calculated phase functions for different combinations of size distribution parameters. The shape of the measurement data is consistent with at least one of the magenta lines, which all represent slightly smaller diameter than AAC set point while differing in underlying GSD. By contrast, discrepancies between measurement data and all cyan or yellow lines go beyond estimated measurement uncertainty. These

findings support the conclusion that measured phase functions are self-consistent across different angles and that they tightly



constrain the retrieved diameter. The small but systematic low bias of retrieved diameters compared to AAC set points could just as plausibly be attributed to a small bias of the AAC.

These results demonstrate the successful validation of the uNeph. The magnitude of the modelled uNeph measurement errors are plausible, although drawing firm conclusions is difficult because uncertainties in the properties of the validation aerosol

translate to considerable uncertainties in the predicted phase functions. In other words, the information content of the phase functions measured by the uNeph for a unimodal aerosol with known RI is so high, that aerosol properties (i.e., $d_m$, $GSD$ and $N_{tot}$) retrieved with a suitable inversion algorithm have uncertainties that are similar or even smaller than the prior knowledge of these properties. This statement is in line with the findings of the uNeph information content analysis presented in Moallemi et al. (2022).

The uNeph measurement accuracy appears to be comparable to that of previous laser imaging type nephelometers. Dolgos and Martins. (2014) estimated errors to be on the level of ~5% for $F_{11}$ and ~0.05 (absolute) for PPF in their laser imaging nephelometer. Ahern et al. (2022) reported a precision of ±2% for $F_{11}(\theta)$ and a positive bias of ~30% for the integrated scattering coefficient obtained by integrating $F_{11}(\theta)$ over $\theta$.

The results for the 200 and 400 nm examples shown in Fig. 6 (left panels) show that the uNeph can provide phase

function measurements between 5˚ to 175˚ under favourable conditions. On the other hand, a portion of the phase functions measured for the 600 nm and 800 nm cases in the ~ sub 35˚ angular range was discarded during the data processing step described in Section 3.2.1. This is the consequence of operating the uNeph in an overly sensitive configuration combined with insufficient dynamic range, which potentially leads to systematic measurement bias when probing large aerosol particles (see Section 3.2.1 for extensive discussion). Unfortunately, the CCD was

accidentally operated without cooling, which affected its dynamic range compared to operation in a cooled state. To investigate how relaxing criteria for filtering proper signals can affect the measurement results, we modified the data processing code to retain the image data acquired with $t_{expo} = 0.1$ s. These data, shown as black dashed lined in Fig. 6, are systematically larger than the predicted values. Therefore, it is considered important to rigorously filter the raw data following the procedure described in Section 3.2.1 to achieve high quality data.

**5.3 uNeph-GRASP retrieval of aerosol properties**

Retrieving aerosol properties from measured phase functions is a central application of aerosol polarimetry. Here we present a first test experiment to demonstrate feasibility of aerosol property retrieval from uNeph measurements. We used the GRASP-algorithm (Dubovik et al., 2014) to solve the inverse problem. GRASP is a versatile and well-established inversion algorithm used in a wide range of aerosol remote sensing applications (Dubovik et al., 2021). GRASP uses a multi-term least squares

minimization in measurement space as the basis for solving the aerosol-light scattering inverse problem, including



consideration of a priori constraints. It allows for using multiple types of measurement inputs, and to retrieve different types of aerosol properties. For our purpose, we tailored the open-source version GRASP OPEN (https://www.grasp-open.com/; last access: 25 May, 2022) to handle the single scattering inverse problem for uNeph phase function data, hereafter referred to as *uNeph-GRASP inversion*.

GRASP is mainly designed for atmospheric applications with relatively wide size distributions. The standard GRASP kernel is a pre-computed lookup-table with a finite resolution on diameter scale. This can result in discretization errors during the retrieval for lognormal size distributions with a GSD smaller than around 1.2, while such errors are negligible for wider size distributions. Therefore, to assess uNeph-GRASP inversion (i.e., to investigate retrieval of aerosol volume size distribution and RI from uNeph-measured $F_{11}$ and $-F_{12}/F_{11}$) a sufficiently broad aerosol size distribution had to be generated. For this
purpose, we generated a broad unimodal DEHS aerosol following the experimental procedure illustrated in Fig. 3c and explained in Section 2.2.

The aerosol was first generated and collected in a tank serving as holding chamber. Then the aerosol was sampled from this tank and probed by the uNeph as well as an SMPS and CPC to provide independent measurements of particle size distributions and number concentrations, respectively. Sampling from the tank led to gradual dilution and concurrent decrease of aerosol
concentration during the uNeph measurement. In order to minimize systematic measurement bias, this concentration drift was accounted for in the uNeph data processing. Specifically, the time-resolved particle number concentration data measured by the CPC were used to compensate the $F_1(\theta)$ and $F_2(\theta)$ data, which were measured by the uNeph at different times (~5 min total measurement time per polarization set point with a 1 h time gap in between).

The measurement space considered in the uNeph-GRASP-inversion either consists of i) $F_{11}$ only or ii) $F_{11}$ and $-F_{11}/F_{12}$. The
DEHS test aerosol consists of an ensemble of homogeneous spherical particles with equal RI and a unimodal size distribution. Therefore, we chose the following two variants of aerosol state space representations for the uNeph-GRASP inversion: i) lognormal volume size distribution representation (state parameters: total volume concentration, $V_{tot}$; geometric mean radius, $r_g$; geometric standard deviation, $GSD$) or ii) binned size distribution representation (state parameters: volume concentrations at each size bin, $V_k = dV/d\log r(r_k)$ for 22 size bins with central radii fixed at positions $r_k$). For both variants, the GRASP default
size range covering particle radii from 0.05 µm to 15 µm was used, and particles were assumed to be spherical with real and imaginary parts of RI allowed to vary in the ranges from 1.35 to 1.7 and $10^{-5}i$ to $0.2i$, respectively. The $GSD$ was allowed to vary across the full available range from 1.2 to 3 for the lognormal size distribution representation. The parameters for imposing size distribution smoothness constraints for the binned size distribution representation were chosen based on the values used by Dubovik et al. (2011) for aerosol property retrieval over a single satellite pixel (difference order = 3 and Lagrange multiplier
= 0.005 for volume size distribution). No further constraints such as, e.g., forcing size distribution tails to zero were applied.



Figure 7 presents the $F_{11}$ and $-F_{12}/F_{11}$ measured by the uNeph for the test sample, together with the fit results for different uNeph-GRASP inversions (*fit* refers to $F_{11}$ and $-F_{12}/F_{11}$ calculated using the GRASP forward model and the inverted aerosol properties). All four uNeph-GRASP inversion configurations result in largely identical fit curves (blue and red colored lines in panels a and b), but for the exception in the angle range ~150° to 170° for the binned configurations, which will be addressed later in this Section. The GRASP fits to the $F_{11}$ match the measurements at the majority of angles (panel a). Discrepancies slightly beyond error margins only occur in the angle ranges from ~95° to ~105° (all retrieval settings) and from ~155° to ~170° (log-normal retrieval settings). The GRASP fits to the $-F_{12}/F_{11}$ function match the uNeph measurements reasonably well (panel b). However, noticeable differences beyond the error margins occur in the angle ranges from ~60 to ~85˚ and from ~110° to ~160°. The exact reasons for these discrepancies for $F_{11}$ and $-F_{12}/F_{11}$ remain elusive as multiple factors can play a role: e.g., residual bias in the compensated aerosol concentration drifts, slightly underestimated measurement errors, or fine structures in the true aerosol size distribution shape that cannot be reproduced by the aerosol size distribution representations implemented in GRASP.

The uNeph-GRASP inversion results for the unimodal DEHS aerosol (i.e., retrieved aerosol volume size distribution and complex RI) are shown in Fig. 8. The volume particle size distributions (VPSD) retrieved with the four uNeph-GRASP inversion variants are in close agreement with each other (red and blue colored lines in panel a), which explains the close match of four corresponding GRASP fits to $F_{11}$ and $-F_{12}/F_{11}$ in Fig. 7. There is a small but clearly discernible difference in the retrieved VPSD from the binned inversions, which have additional minor modes to the right of the main mode. This leads to better agreement between GRASP-fit and measured $F_{11}$ in the angle range of 150-170˚ (Fig. 7a) compared to the other retrievals. However, substantial particle volume in this size range is not expected based on the aerosol generation process, hence, it could be a result of over-fitting measurement bias. The uNeph-GRASP inversion also retrieves both the real and imaginary parts of the RI. These are in very close agreement among the four inversion variants, i.e., maximal absolute differences are as small as 0.018 and ~$3 \cdot 10^{-4}$i for the real and imaginary parts, respectively (colored markers in Fig. 8b and Fig. 8c).

The results discussed so far show that the uNeph-GRASP inversion is robust in the sense that it leads to essentially identical retrieval results for different inversion variants applied to the unimodal test aerosol, which also reproduce the measured $F_{11}$ and $-F_{12}/F_{11}$ reasonably well. As a last step, we compare the retrieval results with independently known or measured properties of the aerosol sample. Validation of retrieved VPSD is done against independent measurements by an SMPS. Figure 8a shows that the mode and width of the size distributions from SMPS (black line) and uNeph-GRASP inversions (coloured lines) are consistent, while the magnitude of the retrieved size distribution is larger than that of the independent measurement. For a quantitative comparison, the integral properties total volume concentration ($V_{tot}$), geometric mean radius ($r_g$) and *GSD* were calculated from the measured and retrieved VPSDs (when not directly delivered as retrieved parameter).



The results listed in Table 2 demonstrate agreement for $V_{tot}$ between SMPS and each retrieval result within 45% or better. This is fair agreement, though outside the range of expected uncertainties of either approach under optimal performance. The reasons for this discrepancy remain elusive. In contrast to $V_{tot}$, the retrieval results for the state parameters $r_g$ and $GSD$ indicate

that the uNeph-GRASP retrieval and the SMPS measurement agree quite well (Table 2). The binned retrievals have the largest $r_g$ and $GSD$, which is caused by the minor tail in the retrieved VPSD at larger diameters. Agreement with the SMPS remains good despite this retrieval artefact. The lognormal retrieval variants provide slightly larger $r_g$ than the SMPS (+10%; 0.275 µm instead of 0.250 µm) and slightly narrower $GSD$ (~1.22 instead of 1.3). This is good agreement, thus validating the uNeph-GRASP inversion for retrieving VPSD width and size for unimodal aerosols.

Excellent agreement between independent knowledge and uNeph-GRASP inversion was also achieved for the RI. The literature value for $n$ (i.e., the real part of the RI) of DEHS (1.455; Pettersson et al. 2004) is virtually identical to the retrieval results (~1.431 to 1.449), whereby the lowest retrieval value originates from the $F_{11}$-only/binned retrieval which has a second mode in the size distribution. The retrievals also correctly return a negligibly small value for $k$, i.e., the imaginary part of RI, ($1·10^{-5}$i to $5·10^{-4}$i), which is perfectly consistent with the fact that DEHS is a non-absorbing liquid with $k$ smaller than $1·10^{-4}$i

(Verhaege et al., 2009).

The accuracy achieved in retrieving aerosol properties for the broad unimodal DEHS aerosol test case can be explained by the high information content of uNeph measurements. Recently, an information content analysis conducted by Moallemi et al. (2022) demonstrated that for unimodal DEHS aerosol test cases, even a polar nephelometer with high angular resolution basic radiometric configuration, i.e., only $F_{11}$ measurements at a single wavelength, can already be quite informative in retrieving

aerosol state parameters. Considering that the level of the uNeph measurement errors for the broad unimodal test aerosol is similar or lower to the base case in the aforementioned information content study, it is not surprising that aerosol properties can be retrieved with high accuracy. Furthermore, the missing benefit in retrieval accuracy of including $-F_{12}/F_{11}$ is not surprising, as this benefit becomes more prominent for more complex aerosols, as shown by Moallemi et al. (2022).

It should be noted that retrieval of $k$ using light scattering measurements is generally more challenging than for other aerosol

properties. Often, accurate absorption retrievals require auxiliary measurements, such as aerosol extinction or absorption (Schuster et al., 2019). The information content study by Moallemi et al. (2022) indicates that the scattering measurement has higher information content for retrieval of $k$ when the aerosol is non-absorbing. This, in combination with limited complexity of the probed aerosol, explains why good results were also achieved by the uNeph-GRASP inversion for retrieval of $k$ for the non-absorbing DEHS aerosol test case.

Overall, the results from this experiment validate the uNeph-GRASP inversion to perform in situ polarimetric measurements and retrieve aerosol properties. Espinosa et al. (2017, 2019) have already shown the applicability of GRASP for aerosol



property retrieval from laser imaging nephelometer measurements. Our results demonstrate that the GRASP inversion can also be applied to measurements from more compact single wavelength laser imaging nephelometers, such as the uNeph.

## 6 Conclusions

This paper introduces a new laser imaging nephelometer, the uNeph, which measures the scattering coefficient as a function of polar angle at two different states of polarization. These measurements are used to derive the absolute scattering phase function, $F_{11}$, and polarized phase function, $-F_{12}/F_{11}$. The instrument design and all key data processing steps are presented. We further discuss all calibration parameters and the calibration process relying on measuring both gases and a PSL aerosol size standard to achieve optimal calibration accuracy.

We constructed an error model and characterized the uncertainties of the parameters driving the overall measurement error. This makes it possible to provide quantitative estimates of measurement error, which depend on actual results such as phase function shape and absolute scattering intensities. Estimated measurement errors mostly are between 5% to 10 % for $F_{11}$ and smaller than ~0.1 (absolute) for $-F_{12}/F_{11}$, while errors can become larger close to the lower limit of detection.

        The uNeph instrument was validated using DEHS mono-disperse aerosol particles with aerodynamic diameters of 200 nm,
400 nm, 600 nm and 800 nm serving as reference. Good agreement was achieved between the measurements and theoretical phase functions predicted for the reference aerosols. The error model provides plausible measurement errors. However, it was not possible to rigorously validate the error model. This is mainly due to the fact that the estimated measurement errors translate to corresponding uncertainties in aerosol property space which are smaller than or comparable to the available independent knowledge of the reference aerosol properties. The results further demonstrate that it is possible to cover an angle range
between ~5° to ~175° for suitable samples. The small detection cell combined with high sensitivity imposes some limitations. For instance, a smaller sensitive volume is more susceptible to statistical fluctuations of the average particle number inside the sensitive volume which can limit temporal resolution at low particle number concentrations. Moreover, signal saturation can limit the maximum detectable particle size, above which systematic bias occurs in parts of the phase function, if operated at comparable sensitivity for homogeneous samples (e.g. particle free air).

Finally, we performed an experiment for testing the combination of the uNeph and the GRASP inversion algorithm for aerosol property retrievals. This was successfully achieved for a broad unimodal DEHS aerosol sample, i.e., retrieved volume concentration, modal size, size distribution width, and complex refractive index agreed within uncertainty with independent measurements and literature data. Generally, these experiments demonstrate the high information content of uNeph measurement data, i.e., radiometric calibration and measurement errors are sufficient for quantitative aerosol retrieval.



Overall, the uNeph was successfully validated. Future improvements will aim at increasing the dynamic range and the number of measurement wavelengths, with the goal to use it in laboratory and field settings for aerosol characterization, as well as for validation and optimization of polarimetric aerosol property retrievals for more complex aerosols.

**Appendix A: Image data reduction**

**A1 Hot pixel removal**

To remove hot pixels, we used the median of multiple dark image samples (3 repeats) that were acquired at an exposure time of 700 s, which is denoted as $\overline{Im}_{D,700s}$. Assuming that the dark current is mostly homogeneous over the image pixels of $\overline{Im}_{D,700s}$ we calculated the median and standard deviation of all the pixel values of $\overline{Im}_{D,700s}$ which were used in Eq. A1 to define a hot pixel signal threshold ($L_{hot}$):

$$L_{hot} = \text{median}(\overline{Im}_{D,700s}) + 1.5 \times \text{std}(\overline{Im}_{D,700s}), \tag{A1}$$

Any pixel in $\overline{Im}_{D,700s}$ with signal larger than the threshold value is considered a hot pixel and is removed from the image prior to any further signal processing steps. Figure S3 shows the hot pixels in a sample dark image. It should be noted that this process was also tested with dark image samples acquired at exposure times of 215 s and 464 s and the detected pixels were quite similar to the hot pixels obtained by sample images with exposure time of 700 s. Specifically, the 215 s and 462 s test cases detected 2 and 1 hot pixels less than the 700 s samples, respectively.

**A2 Dark signal characterization**

    The CCD image data contain a dark signal contribution which also occurs in the absence of any illumination. The dark signal (*DS*) has two systematic components, a constant positive bias (B) and dark current proportional to exposure time. These can be described with a linear equation:

$$DS = G_{DC} \times t_{expo} + B, \tag{A2}$$

where $G_{DC}$ is the proportionality constant for the dark current part. The dark signal also contains superimposed random noise, which is not captured by Eq. A2. In our analysis we used two different $t_{expo}$, of 4.64 s and 700 s to characterize the *DS*. The *DS* may vary between pixels. Accordingly, the two constants can either be determined for each pixel, or for each angle (after integration over beam cross-section). We considered the signal from dark images collected at the short $t_{expo}$ of 4.64 s (30 sample images) to mainly consist of *B*. Therefore, the mean values of dark images at $t_{expo}$ of 4.64 s were used to calculate *B*.

Subsequently, we used the mean signal of dark sample images acquired at the long $t_{expo}$ of 700 s to obtain $G_{DC}$.

    To determine the robustness of the dark signal estimation we applied the dark signal correction on a series of dark image signals acquired over $t_{expo}$ ranging from 0.1 s to 464 s. Residuals of the dark signal compensation, that is the difference between



estimated dark signal and actual dark image signals, were integrated over scatter signal bounds for measured scattering angles (Section 3.1.3), and are shown over different $t_{expo}$ in Fig. S4. The results show that the dark residual signal, $\Xi_{DRS}(\theta)$, are ~

$\pm200$ a.u. for the uncooled CCD case over the scattering angles for the sample acquired at $t_{expo}$ below 215 s, while for the dark image sample at $t_{expo} = 464$ s, $|\Xi_{DRS}(\theta)|$ as large as 500 a.u. were also observed. The results further suggest that for the CCD cooled case the $\Xi_{DRS}(\theta)$ are ~ $\pm200$ a.u. for all the tested exposure times. Valid light scattering data near the lower limit of detection can only be achieved if the interference of dark signal residuals is small, i.e., if the contribution of light scattering to the CCD signal substantially exceeds the dark signal residuals. The uNeph data presented in this manuscript were all acquired

with uncooled CCD.

**A3 Scattering angle calibration procedure**

The relationship between scattering angle and image pixels is determined through the scattering angle calibration. To conduct the scattering angle calibration a multi-stepper motor mechanism, which is referred to as the *three-dimensional (3D) position probe*, was employed. Figure S5a shows the three-dimensional (3D) position probe which is made up of three orthogonal step

motors traversing stages which are intended to travel along the *xyz* coordinate axes according to the coordinate system depicted in Fig. S5a. A probing arm with a pinhead at the end of the arm was mounted on the traversing stage that travels along the *x* axis. The function of the 3D positional probe is to probe the spatial location of the optical objective pinhole and different locations along the path of forward and backward beams. The stepper motors used in the 3D positional probe traverse a distance of 0.025 mm per step.

To conduct the scattering angle calibration, the top cap of the scattering chamber is removed. The 3D positional probe is then mounted on top of the uNeph instrument such that the probe arm has access to the scattering chamber. The initial step for conducting the angular calibration is the identification of the objective pinhole coordinate. To do so, the pinhole was moved from a reference position (the origin coordinate) and was carefully displaced with step motor movements until the probe pinhead reaches the location of the objective pinhole, which we define as point *P*. The steps taken by the motors from the

origin point was recorded and based on that the coordinates of the objective pinhole is specified relative to the origin point. Subsequently, a similar approach is employed to obtain the coordinates of the centre of the laser beams in the *yz* plane. The next step in the calibration process involves placing the probe pinhead at different locations along the laser beam central axis and within the field of view of the objective pinhole lens. The coordinate of a point along the laser beam centre axis is denoted as point *S*. The polar scattering angle then becomes:

$\theta = \arccos(\frac{\overrightarrow{SP} \cdot \hat{\imath}}{\|\overrightarrow{SP}\|})$                                                                                                       (A3)

In Eq. A3, *P* is the objective pinhole position, $\overrightarrow{SP}$ is the vector connecting point *S* to *P* with vector length of $\|\overrightarrow{SP}\|$, and $\hat{\imath}$ is the unit vector along the *x* axis (laser beam axis). Figure S5b shows an example scheme where scattering angle geometry is depicted for a case where the probe pinhead is placed at a given location *S*. When the probe pinhead is located at a given



position (e.g., point S), the light reflected of the pinhead will be detected by the imaging unit and generating a bright spot in a
sample image capture by the CCD camera. During the calibration process the probe was placed in 28 different locations along
the laser beam centre axis of each beam (forward and backward beams). Once placed at each of these locations a picture was
taken of the reflected spot of the probe pinhead. Figure S6a shows an example of the pinhead spot detected by the CCD at a
single location and Fig. S6b shows a composite image of the combination of multiple pinhead spots images that were taken
during the scattering angle calibration process. Thus far, this process provides pairs of polar angles (Eq. A3) and spots on the
CCD image with finite width. As a last step, an exact coordinate on the CCD image (i.e., single representative pixel) is assigned
to each of these spots by calculating its centre of mass (red dots in Fig. S6).

While the pixel-angle information provided by the angular calibration are useful, they are quite limited and the angular
difference ($\Delta\theta$) between most of the adjacent calibration pixels are larger than 1˚. Therefore, further processing is required to
obtain pixel-angle information with angular resolution of 1˚. To refine the pixel-angle data, the angular calibration points at
each of the beam segments (forward/backward) were used to generate a second-degree polynomial fit which takes angles as
input and returns pixel coordinates. These fits can be used to generate a list of refined coordinates corresponding to scattering
angles ranging from 3-90˚ (forward beam) to 90-177˚ (backward beam) with angular resolution of 1˚.

**A4 Image transformation and signal integration limits**

Sections 3.1.2 and A3 described how to obtain the red fit curve in Fig. 2b which is a parametrization of pixels coordinates as
a function of $\theta$. The next goal is to extract a pixel array representing laser beam image cross-sections for each polar angle ($\theta_i$).
For this purpose, lines perpendicular to this fit were obtained as:

$$y - y_{CL}(\theta_i) = -\frac{1}{m(\theta_i)}(x - x_{CL}(\theta_i))$$  (A4)

In Eq. A4, $y_{CL}(\theta_i)$ and $x_{CL}(\theta_i)$ are the $x$ and $y$ coordinates of points along the laser beam centre line chosen with an angular
resolution of 1°, and $m(\theta_i)$ is the gradient of the fitted curve at angle $\theta_i$. We then considered the pixels closest to these
perpendicular lines to extract the beam cross sections for each ($\theta_i$). Figure S7 shows the transformed image with beam cross
sections as a function of $\theta$. 120 pixels were extracted for each angle such that the full beam cross-section is included in the
transformed image.

The next step involves defining the integration limits for each angle, inside which the signal is considered for further analyses.
We used the $CO_2$ sample images under perpendicular polarization condition and chose as boundaries those pixels where the
signal dropped to 10% of the maximum signal at the centre of the cross section (Fig. S7). Repetitions of boundary pixel



identification performed over a period of approximately one month revealed stability of instrument geometry and optics such that we chose to use the median result for all further data analyses.

**A5 Evaluation of uNeph measurement error components**

The error component $\sigma_{BG,l}$ depends on precision of background subtraction. We estimate a relative error in $\xi_{BG}$ of around ±3%. This is based on observed stability and random noise in measured air background data (Figs S14-S16). The corresponding error $\sigma_{BG,l}$ is obtained as the difference between perturbed and unperturbed measurement results, where the perturbed measurement results are calculated based on perturbing $\xi_{BG}$ by ±3% in the air background subtraction step. This directly provides $\sigma_{BG,F_1}$ and $\sigma_{BG,F_2}$ for positive and negative perturbations. The air background values of polarization states 1 and 2

shown in Fig. S14 and Fig. S15, respectively, exhibit a high covariance. Therefore, we make the simplifying assumption that air background error is fully covariant for these two measurements. Hence, the perturbed $F_{11}$ is to be calculated by inserting perturbed $F_1$ and $F_2$ into Eq. 3, whereby the air background was perturbed with identical sign. $-F_{12}/F_{11}$ is perturbed equivalently to $F_{11}$ to account for error covariance. The effect of the error component $\sigma_{BG,l}$ can be highly variable depending on the ratio of particle signal to air signal. To demonstrate this, we conducted error analysis on 250 nm mono-disperse DEHS aerosol particles

that were measured at two particle number concentration levels of 3145 cm$^{-3}$ (high) and 69 cm$^{-3}$ (low). Figure S22 shows the measured phase function with estimated total error and contributing error components for the high concentration experiment. $F_1$, $F_2$ and $F_{11}$ of the aerosol (red curves in panels a-c) remain well above 3% of the air background (dashed blue lines) at all angles. Therefore, the contribution of $\sigma_{BG,l}$ to error (magenta lines in panels e-h) is estimated to remain below ~5% for all angles. By contrast, $F_1$, $F_2$ and $F_{11}$ of the aerosol (red curves in Fig. S23a-c) are comparable or smaller than 3% of the air

background (dashed blue lines) for backward scattering ($\theta > $ ~110°). Accordingly, the contribution of $\sigma_{BG,l}$ to error (magenta lines in Fig. S23e-h) is estimated to be large in this angle range. Indeed, the phase functions measured for the high concentration case are smooth and follow a fitted Mie curve across all scattering angles, while the measurement of the low concentration example is noisier and in poorer agreement with the fitted Mie curve at backward scattering angles (note, Mie curve fitting is discussed in Section 5.2).

The error component $\sigma_{BG,l}$ was further assessed on measurements acquired with 600 nm PSL aerosols and is shown in Fig. 5. $F_2$ for this aerosol (black line) has distinct features, i.e., it drops to very small values at, e.g., ~105° and ~155°, which is typical for monodisperse aerosols in this size range. Accordingly, the error model predicts distinct peaks for $\sigma_{BG,F_2}$ at these angles (magenta line in Fig. 5f). The relative error from $\sigma_{BG,F_2}$ reaches up to ~25% and ~15% at 105° and 155°, respectively. This is not the case for $F_1$ for which the aerosol signal remains clearly above air background at all angles, thus resulting in negligible

error from background subtraction (magenta line in Fig. 5e). Propagating background subtraction errors in $F_1$ and $F_2$ to errors in $F_{11}$ and $-F_{12}/F_{11}$ results in $\sigma_{BG,F_{11}}(\theta) \leq 3\%$ and $\sigma_{BG,\frac{-F_{12}}{F_{11}}}(\theta) \leq 0.03$. The peaks at angles around 105° and 155° in error of $F_2$ from background subtraction are heavily dampened, but they remain discernible and the dominant source of error at these



angles for $-F_{12}/F_{11}$ (magenta line in panel Fig. 5h). Overall, the error analysis results for the low and high concentration 250 nm DEHS and the 600 nm PSL examples show that air background subtraction becomes a major source of error whenever aerosol

signal becomes too small compared to air background, and further indicate that the error model plausibly reproduces this effect.

The error component $\sigma_{\theta,l}$ depends on angle calibration accuracy. Angle calibration errors mainly depend on accuracy of the measured pinhole location (Section 4.3). Therefore, to estimate angle calibration errors, we perturbed the pinhole position within its estimated uncertainty range. To obtain the angular perturbation ($\theta\pm\delta\theta$), the (optimized) optical pinhole location was perturbed by $\pm0.1$ mm in x, y and z, creating 27 perturbed cases. The perturbed pinhole configuration and corresponding angles

with the largest differences (positive or negative) to the optimal angles were identified and employed in the error analysis. Resulting angle calibration errors, $\Delta\theta$, i.e., the maximal difference between perturbed and unperturbed angles, varied between 0.07° and 1.2° over the full angle range (Fig. S21). The error component $\sigma_{\theta,l}$ was determined as the difference between the uNeph measurement results obtained using either the unperturbed or perturbed angles in the data processing chain. This calculation is somewhat more complicated than, e.g., the air BG perturbation calculation for the following reason: if perturbed

angles are assumed to be *true*, then the gain calibration factors $G_1$ and $G_2$ derived with unperturbed angles are biased. Therefore, calculation of perturbed phase functions involves multiple changes to achieve a consistent assessment of errors. First, the angle scale is replaced by perturbed angles. Second, the calibration data are re-evaluated to determine perturbed gains. Third, processing of uNeph measurement data for the PSL sample is repeated using these perturbed angles and gain calibration constants. Potential errors in angle calibration are identical for both laser polarization set points. Therefore, angle perturbations

$\Delta\theta$ are assumed to be fully covariant when using perturbed $F_1$ and $F_2$ measurements to calculate perturbed $F_{11}$ and $-F_{12}/F_{11}$ (equivalently to handling covariance in BG subtraction error). The error component $\sigma_{\theta,l}$ for the 600 nm PSL aerosol test case is shown in Fig. 5. The blue lines in panels e and f demonstrate that $\sigma_{\theta,l}$ remains small for $F_1$ and $F_2$ at angles at which their gradients (i.e., derivative by $\theta$) remain small. By contrast, regions with large phase function gradients result in considerable $\sigma_{\theta,l}$ of up to 35% in panel f. The direct relation between $F_2$ gradient and $\sigma_{\theta,l}$ is clearly seen near the local minima of $F_2$, for

example across the angle range from 135°-170° (panels b and f). The error has two peaks at ~147° and ~160°, which are separated by a sharp drop in error at the local minimum of $F_2$ (at 155°). Propagating $\sigma_{\theta,F_1}$ and $\sigma_{\theta,F_2}$ to $\sigma_{\theta,F_{11}}$ has a dampening effect, essentially because high gradients in $F_1$ and $F_2$ occur at different angles, such that $\sigma_{\theta,F_{11}}$ remains below 10%, except for slight exceedance in the angle range 40° to 60°. Similarly, errors in $\sigma_{\theta,\frac{-F_{12}}{F_{11}}}$ remain below 0.1 (absolute) for this test aerosol example (Fig. 5, panel h).

The error component $\sigma_{q,l}$ depends on uncertainty in the laser polarization state parameters $q_1$ and $q_2$, which appear in Eq. 3. We use an uncertainty of $\Delta q \approx \pm0.05$ for calculating $\sigma_{q,l}$. This estimate for $\Delta q$ is based on the fact that perturbing $q_1$ and $q_2$ by this much results in clearly discernible systematic deviation between theoretical and measured $F_1$ for argon gas at scattering angles around $\theta = 90°$ (Fig. S19). We determined $\sigma_{q,F_1}$ and $\sigma_{q,F_2}$ by subtracting uNeph results obtained with using perturbed $q$





values from the unperturbed measurement results. The gain calibration constants $G_1$ and $G_2$ were also re-evaluated to ensure
consistency, analogously to the error calculation for perturbed $\theta$. Potential calibration biases in $q_1$ and in $q_2$ are expected to be
independent of each other, hence, corresponding $\sigma_{q,F_1}$ and $\sigma_{q,F_2}$ are also expected to be independent of each other.
Consequently, standard equations for error propagation of independent measurement errors were used to infer $\sigma_{q,F_{11}}$ and $\sigma_{q,\frac{-F12}{F11}}$
from $\sigma_{q,F_1}$ and $\sigma_{q,F_2}$. Note, this step differs from the corresponding step in the calculation of $\sigma_{\theta,l}$, or $\sigma_{\mathrm{BG},l}$, where error
covariance had to be considered. The error component $\sigma_{q,l}$ for the 600 nm PSL aerosol test case is also shown in Fig. 5. The
error $\sigma_{q,F_1}$ (red lines) is small for extreme forward angles (near $\theta = 0°$) and backward angles (near $\theta = 180°$), and is increasingly
large as $\theta$ approaches 90°, where it exceeds 10%. High errors in $\sigma_{q,F_1}$ near 90° are caused by the gain calibration step.
Calculation of $G_1$, as described in Section 4.2, is very sensitive to bias in $q_1$. The cause of this effect is that gases have a much
higher partial scattering cross section for perpendicular linearly polarized light than for parallel linearly polarized light at
scattering angles near 90°. Conversely, the error $\sigma_{q,F_2}$ remains very small at all angles for the opposite reason (panel f in Fig. 5).
Panel g shows that the error $\sigma_{q,F_{11}}$ in $F_{11}$ is similar to the error in $F_1$ at all angles where $F_1$ is much greater than $F_2$. Conversely,
the error in $\sigma_{q,F_{11}}$ is dampened, compared to $\sigma_{q,F_1}$, at angles where $F_2$ is similar or larger than $F_1$. These effects are nicely seen
when, e.g., comparing $\sigma_{q,F_{11}}$ at the angles ~85° and ~95° with corresponding errors $\sigma_{q,F_1}$. Also, the error in the polarized phase
function caused by uncertainty of $q$ values, $\sigma_{q,\frac{-F12}{F11}}$, is most pronounced at angles around 90°, where it reaches maximal
absolute values of almost 0.1 (panel h in Fig. 5).

The precision component $\sigma_{p,l}$ in the error model is introduced to account for random contributions to measurement error which
are not accounted for in the other error components. We already showed that compensated signals of particle free air samples,
$\xi_{\mathrm{air}}$, vary by about ±3% over a period of two weeks. This variability is attributed to variations in uNeph sensitivity and other
random noise (Figs S14-S16). By assuming comparable random variability in $\xi_{\mathrm{aerosol}}$, we estimate ~3% relative error for the
precision component of measurement error in $F_1$ and $F_2$. This is a rather low level of random noise, however, the assessment
of gain calibration variability presented in Section 4.2 supports plausibility of this error estimate, at least for sufficiently high
signal levels. Note, larger random noise is expected to occur when fluctuations in detected particle number become relevant
as demonstrated in Fig. S13 and discussed in Section 3.2.1. The random error components $\sigma_{\varepsilon,F_1}$ and $\sigma_{\varepsilon,F_2}$ are assumed to be
independent of each other, hence, standard equations for propagation of independent errors were used to infer $\sigma_{\varepsilon,F_{11}}$ and $\sigma_{\varepsilon,\frac{-F12}{F11}}$
from $\sigma_{\varepsilon,F_1}$ and $\sigma_{\varepsilon,F_2}$. The error component $\sigma_{\varepsilon,l}$ for the 600 nm PSL aerosol test case is also shown in Fig. 5. The cyan lines in
panels e and f of Fig. 5 reflect the fixed random noise $\sigma_{\varepsilon,F_1}$ and $\sigma_{\varepsilon,F_2}$ directly imposed on $F_1$ and $F_2$. Error propagation leads to
$3\% < \frac{\sigma_{p,F_{11}}}{F_{11}} < \sqrt{2} \cdot 3\%$, and the error $\sigma_{\varepsilon,\frac{-F12}{F11}}$ always remains below 0.05 (absolute).



**Acknowledgements**

Financial support was received from MeteoSwiss through a science project in the framework of the Swiss contribution to the global atmosphere watch programme (GAW-CH) and from the Swiss National Science Foundation (BISAR project; SNSF

grant no. 200021_204823). The authors acknowledge Gergely Dolgos for his contributions to the initial study design and proposal writing and Nicolas Bukowiecki for contributions to proposal writing. The authors acknowledge Oleg Dubovik, Tatyana Lapyonok, Anton Lopatin, and David Fuertes for the support they provided for running GRASP-OPEN.

**Data availability**

The original contributions presented in this study are included in the article and attached supplementary information. The

specific data from this study will be made publicly available on Zenodo if the manuscript is accepted for publication.

**Author contributions**

MGB conceptualized the study and raised the funding. PG lead the instrument design in exchange with MGB, AM and BTB. AM and PG wrote the data acquisition and data analyses software. AM performed the experiments with advice from PG, BTB, RLM and MGB. AM performed the data analyses and theoretical calculations with advice and contributions from RLM, MGB,

and BB. MGB secured the funding for this project. MGB and RLM supervised this project. The original draft was prepared by AM and MGB, and all the co-authors contributed in the writing, revision and editing of the paper.

**Competing interests**

The contact author has declared that none of the authors has any competing interests.

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





**Tables**

**Table 1 Validating absolute values of measured phase functions using PSL size standards and independent number concentration measurement from a CPC.**

| PSL diameter | Nominal Polarization Set point | Particle number concentration | | | |
|---|---|---|---|---|---|
| | | CPC* | uNeph-derived ** median (q25, q75) | uNeph-derived estimated CV*** | Bias of uNeph-derived (uNeph − CPC)/CPC |
| nm | - | cm⁻³ | cm⁻³ | % | % |
| 600 | $F_1$ (parallel) | 103 ± 12 | 98 (96, 102) | 4% | -4.6% |
| 600 | $F_2$ (perpendicular) | 110± 12 | 113 (105, 123) | 12% | 2.7% |
| 240 | $F_1$ (parallel) | 1587± 36 | 1596 (1557, 1671) | 5% | 0.6% |
| 240 | $F_2$ (perpendicular) | 1628± 39 | 1608 (1589, 1635) | 2% | -1.2% |

*Mean CPC measurements with standard deviation as error values (CPC uncertainty: ~±10%)
**Requires Mie theory constrained with reported diameter and CV of PSL size standards. The number concentration was independently derived from the data points at each angle. Here we report statistics of the results from all measured angles.
***The coefficient of variation is estimated using the formula (0.741×inter-quartile range)/median, which is insensitive to outliers.

**Table 2 Size distribution parameters (geometric mean radius, $r_g$; geometric standard deviation, *GSD*; total volume concentration,**
**$V_{tot}$) of the broad aerosol test case obtained from SMPS measurements and retrieved from uNeph-GRASP inversions.**

| | | $r_g$ (μm) | *GSD* (-) | $V_{tot}$ (μm³/cc) |
|---|---|---|---|---|
| **SMPS measurement (mean)** | | 0.25 | 1.30 | 19.7 |
| **uNeph-GRASP inversions** | **Binned with $F_{11}$** | 0.30 | 1.41 | 28.9 |
| | **Binned with $F_{11}$ & - $F_{12}/F_{11}$** | 0.32 | 1.58 | 28.1 |
| | **Lognormal with $F_{11}$** | 0.28 | 1.23 | 26.3 |
| | **Lognormal with $F_{11}$ & - $F_{12}/F_{11}$** | 0.28 | 1.22 | 25.4 |



**Figures**

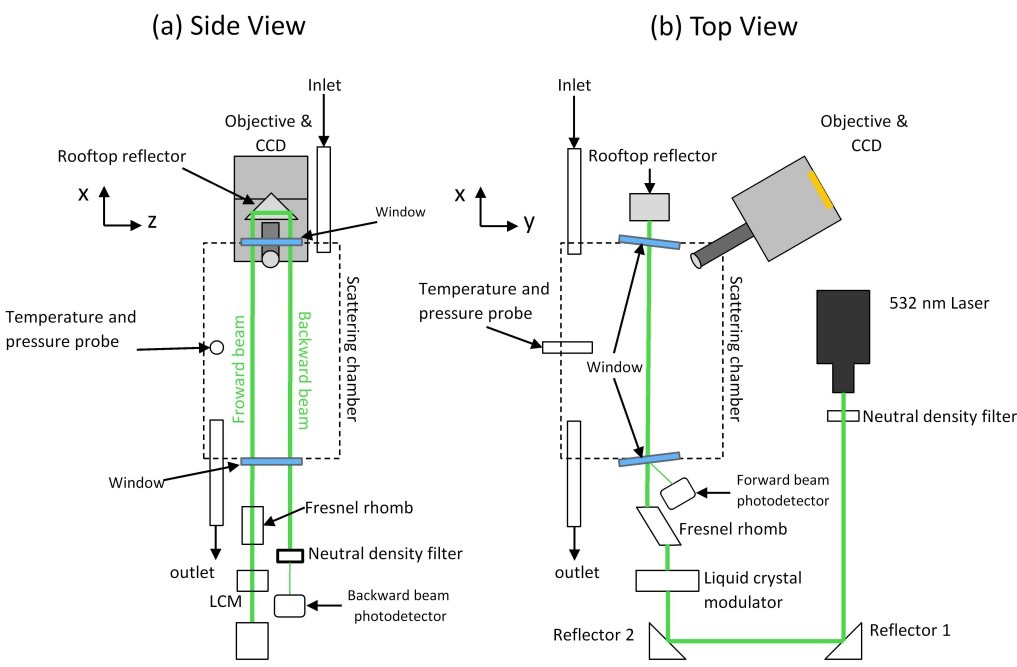

**Figure 1. (a) Side view and (b) top view schematics of the uNeph instrument (the drawing is not to scale).**








**Figure 2. (a) Example of a raw sample image of particle-free air acquired at perpendicular polarization with exposure time of 215 s. (b) the angle calibration fit lines (red) and an example cross-section line (blue) indicating pixels corresponding to scattering angle θ = 40˚. (c) Light scattering image after transformation to angle-pixel coordinates. (d) Cross-sectional signal for the example θ = 40˚. (e) The integrated light scattering signal, Ξ, over measured scattering angles, θ, for a particle-free air sample.**




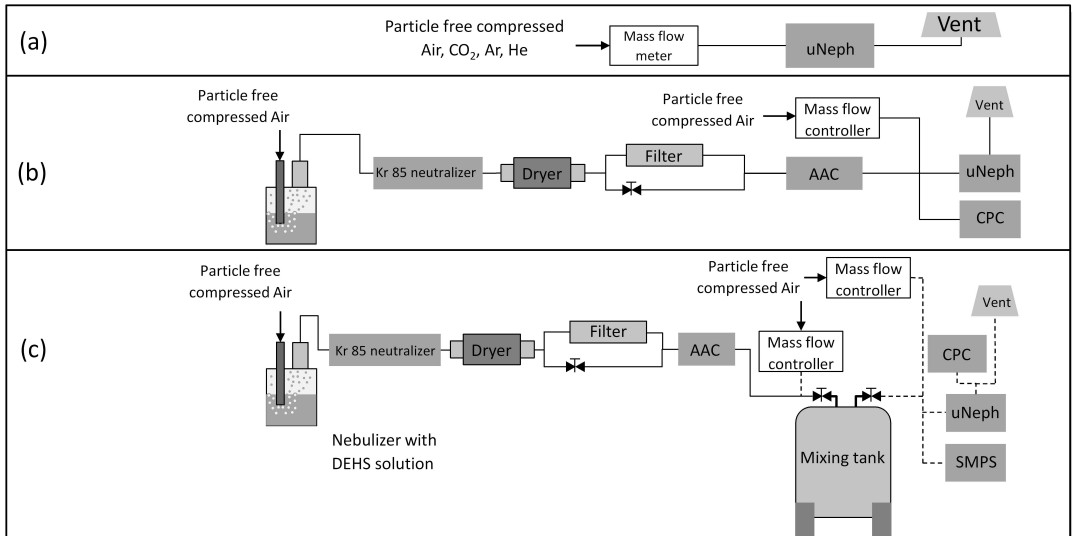

**Figure 3. Schematics of experimental setup for probing (a) gases, (b) quasi mono-disperse aerosol, (c) broad unimodal aerosol. The
aerosol generation step (solid lines) and the sampling step (dashed lines) shown in panel c) were performed in sequence as
described in the main text.**

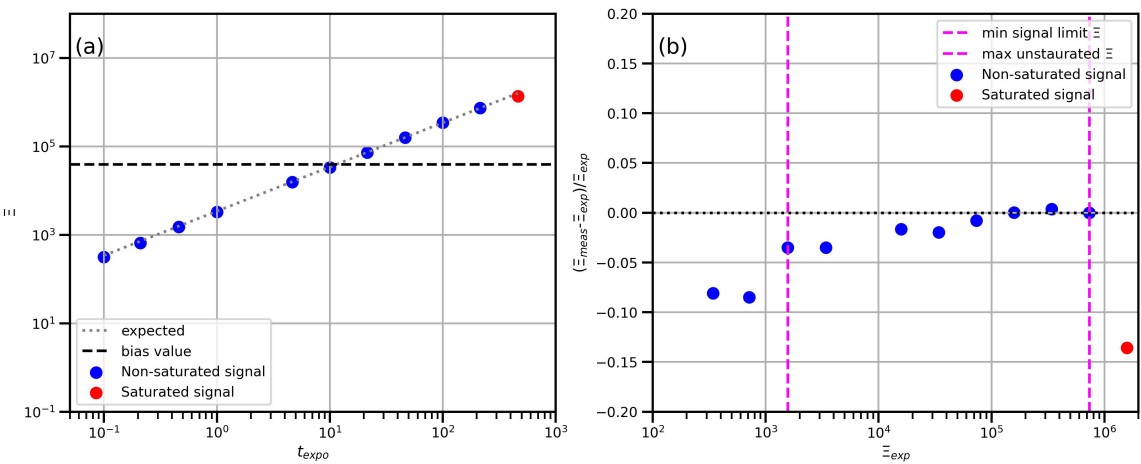

**Figure 4. (a) shows $\Xi(\theta=50°)$ vs $t_{expo}$ for air sample at polarization state 1 collected over different exposure times with CCD cooling
turned on. (b) shows the error of $\Xi_{meas}$ relative to $\Xi_{exp}$ as a function of $\Xi_{exp}$.**



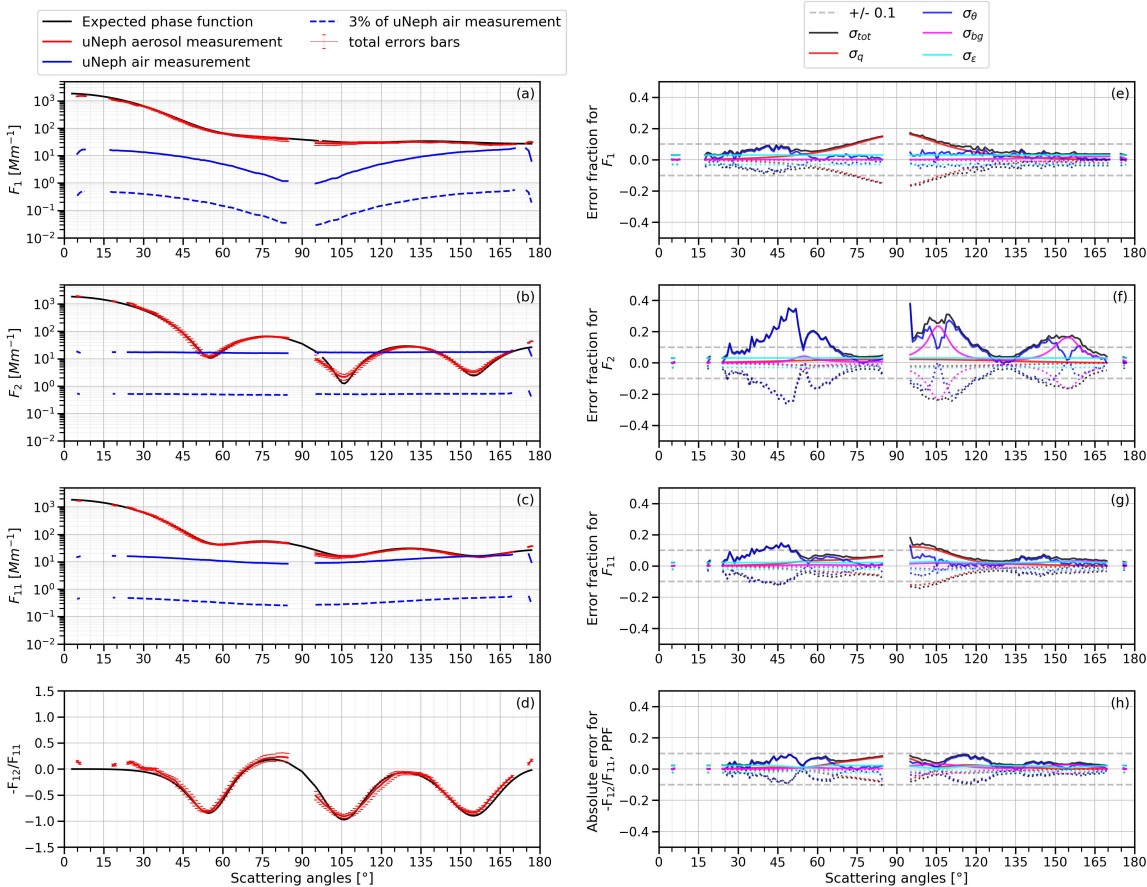

**Figure 5 (a, b, c, d) are the angular light scattering measurements with total errors for the 600 nm PSL aerosol particles (red lines). The black lines are the expected angular measurements that were calculated using Mie theory according to the description in Section 3.5.3. Air background phase functions are also included (blue lines). (e, f, g,) present estimated measurement errors in relative terms for $F_1$, $F_2$, and $F_{11}$, respectively. (h) presents estimated contributions to error of $-F_{12}/F_{11}$ in absolute terms.**



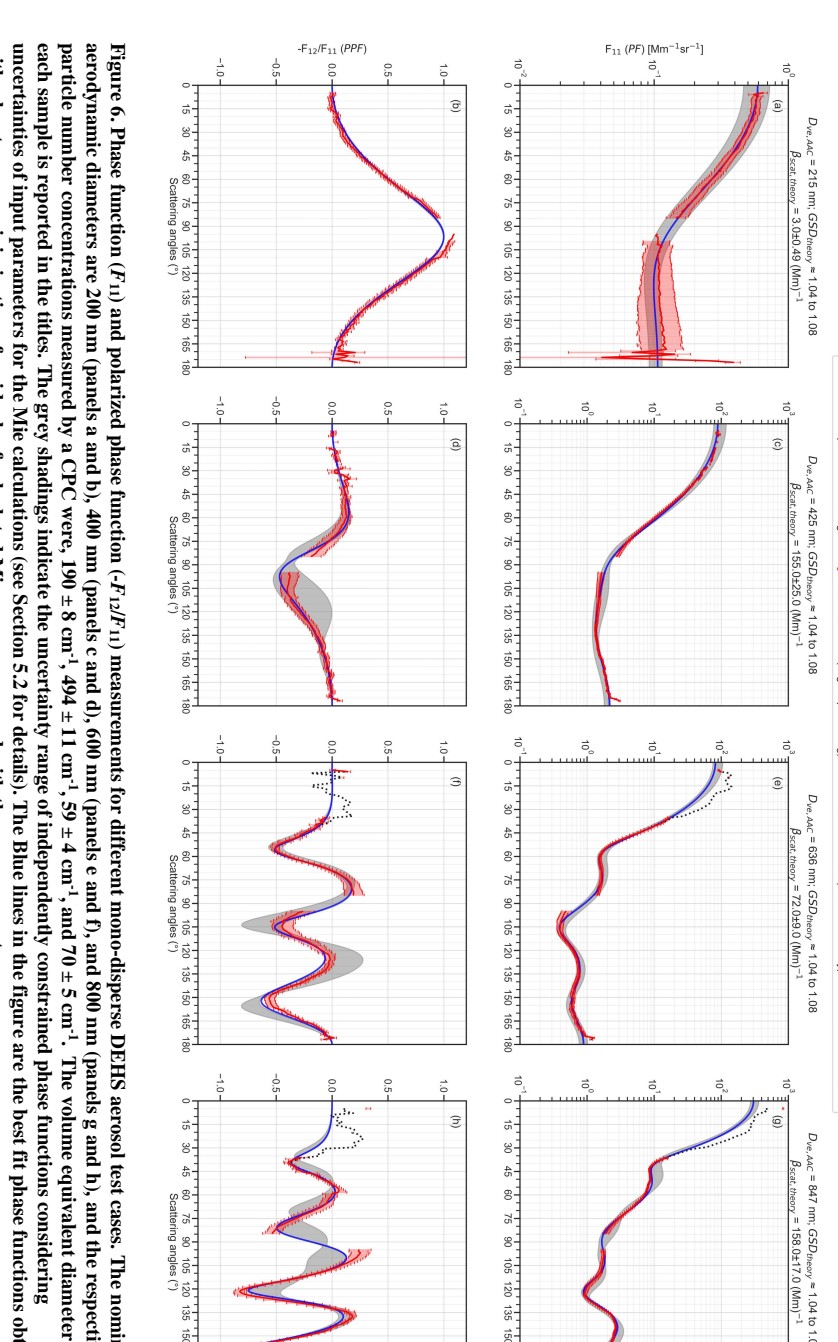

**Figure 6.** Phase function ($F_{11}$) and polarized phase function (-$F_{12}/F_{11}$) measurements for different mono-disperse DEHS aerosol test cases. The nominal aerodynamic diameters are 200 nm (panels a and b), 400 nm (panels c and d), 600 nm (panels e and f), and 800 nm (panels g and h), and the respective particle number concentrations measured by a CPC were, 190 ± 8 cm⁻¹, 494 ± 11 cm⁻¹, 59 ± 4 cm⁻¹, and 70 ± 5 cm⁻¹. The volume equivalent diameter for each sample is reported in the titles. The grey shadings indicate the uncertainty range of independently constrained phase functions considering uncertainties of input parameters for the Mie calculations (see Section 5.2 for details). The Blue lines in the figure are the best fit phase functions obtained with a least square minimization of residuals of calculated Mie curves compared with the measurement.



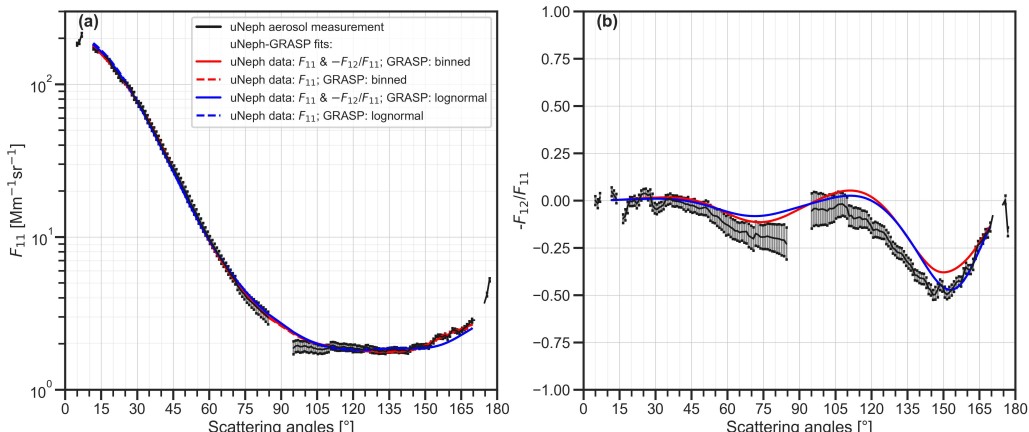

1000

**Figure 7 Phase function ($F_{11}$) and polarized phase function (-$F_{12}$/$F_{11}$) measurements and retrievals by GRASP for the broad unimodal DEHS test case.**

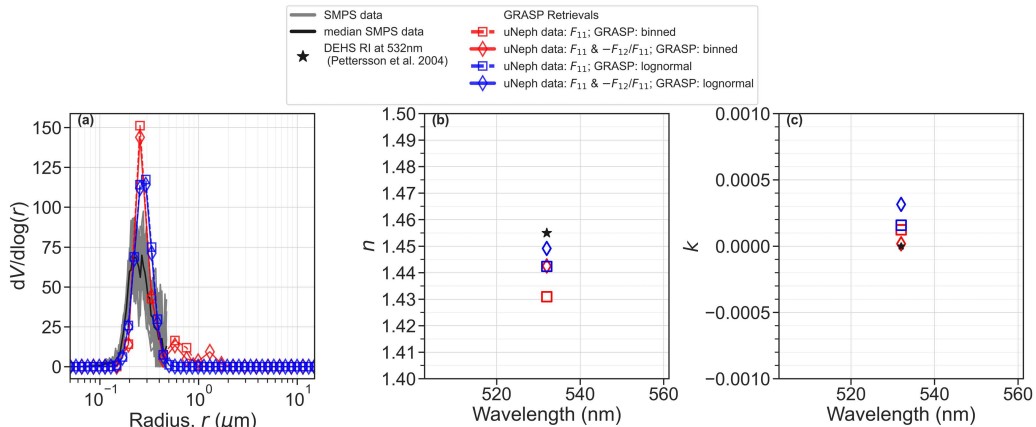

1005

**Figure 8 (a) volume size distribution measurement by SMPS and retrievals provided by applying GRASP on uNeph measurements. (b, c) Retrieval of real (*n*) and imaginary (*k*) parts of refractive index (RI) by applying GRASP on uNeph measurements.**

