# Peer review of "Concept, absolute calibration and validation of a new, bench-top laser imaging polar nephelometer"

_EGUsphere, 2023_

## Author Comment (AC2)

We thank the editor and reviewers for the handling and commenting our manuscript. Please find below our point-by-point responses to each of the reviewers' comments, including the modifications we have made to the manuscript. Reviewer comments are in black text and our responses are given in blue text, and excerpts of the revised manuscript are given in *blue italic text*.

**Reviewer #1:**

General:

This manuscript presents a newly developed polar nephelometer for aerosol phase function characterisation, along with comprehensive calibration discussion and quantitative error analysis. While other groups have demonstrated polar nephelometry with similar configurations, the novelty of this instrument is its reduced size. The authors provide an extremely detailed and rigorous discussion of the data analysis, including the main sources of uncertainty in the data inversion procedures. The performance of the nephelometer and predicted errors are demonstrated via laboratory experiments using monodisperse and unimodally distributed non-absorbing aerosol with known optical properties. While it would have been interesting to see an exploration of more novel materials or more significant instrumentation developments, the strength of this manuscript is in the thorough analysis. I especially appreciate all the detail discussed in the appendix and shown in Supplementary figures. As such, it is a valuable and excellent piece of research that is highly recommended for publication in AMT after a few minor revisions.

Specific comments:

L95: One of the novel features of the uNeph, compared to other polar nephelometers, is its relatively small size. It would be worth reiterating this advantage elsewhere, e.g. the conclusion section.

The conclusions have been updated to reiterate the pros and cons of downsizing the laser imaging uNeph:
*"Overall, the uNeph was successfully validated and our results shows the applicability of a downsized laser imaging nephelometer. Downsizing the instrument posed challenges which result in slower measurements of aerosol samples containing large particles present in small number. At the same time, dimensions of the uNeph have been considerably reduced compared to previous laser imaging nephelometers. This demonstrates that operation of such instruments on e.g. unmanned aerial vehicles is achievable. Such increased flexibility enables acquisition of in situ aerosol polarimetry data sets, which can greatly benefit the remote sensing community for validating and improving existing retrieval algorithms. Future improvements to the uNeph will aim at increasing the dynamic range and the number of measurement wavelengths, with the goal to use it in laboratory and field settings for aerosol characterization, as well as for validation and optimization of polarimetric aerosol property retrievals for more complex aerosols."*

L97-109: Details on the optical system are scant. It would be useful to provide more details on:

- Type of laser and operating mode (i.e. pulsed or cw)
- Whether beam was collimated and estimated beam size within the measurement chamber

- Range of optical densities for ND filters used
- Type of photodetector

In Section 2.1 we added more detail to clarify and address all these points:
*"A solid-state continuous wave laser provides linearly polarized light at 532 nm (~200 mW). The laser beam is collimated with a Gaussian size around 1 mm and vignetted with apertures to slightly cut it down to ~0.5−1.0 mm. Exchangeable neutral density (ND) filters are utilized to reduce the laser intensity in order to adjust instrument sensitivity to different levels. We particularly used three different optical density filters throughout the experiments (Thorlabs, Inc., Newton, NJ, USA, models NE10A, NE15A, NE20A, with optical density levels of 1, 1.5 and 2.0, respectively)."*

and

*"The input window of the scattering chamber is placed at angle and the small reflected part, not blocked by the AR-coating, is used to measure the input laser power (forward beam) with a photodetector (Si free-space amplified photodetectors, Thorlabs, Inc., Newton, NJ, USA, model PDA100A2)."*

L111: Based on Figure 1, it would appear that the camera lens forms part of the chamber seal, and the rooftop reflector is outside a window – is that correct?

Yes, this is correct.

L117-121: Can you clarify the dimensions of the region of interest within the camera pixel array that was used, and whether pixels were binned at all? Despite the large pixel array indicated in Section 2.1, Figure S3 indicates a much smaller array.

When acquiring the images, the pixels were not grouped, nor was this done during conversion from pixel-pixel to angle-pixel coordinates. It should be noted that the result shown in Figure S3 already is transformed to angle-pixel coordinates. During transformation, only a subset of the pixel data is extracted, thus explaining the smaller number of pixels arrays in Figure S3. The pixel-pixel to pixel-angle transformation and signal boundary specification are explained in the Section 3.1 and Fig. 2. Minor edits were made in Section 3.1 to address this referee comment. Part of the referee's question may have arisen from the fact that Fig. 2 was not properly rendered in the document evaluated by reviewer 1 (see separate comment).

L120: More details about the objective (lens) would be helpful, e.g. camera position relative to the focal length, the field-of-view…

We added the following details in Section 2.1 regarding the camera specifications, settings and position:

*"[...]One design element of the uNeph is the use of a wide field of view pinhole lens in the camera objective, which enables the instrument to be downsized. The camera objective is the Marshall V-PL25CS-12 (discontinued) with a pinhole size of ~ 2 mm, a focal length of 2.5mm and aperture of F2.8. The field of view (diameter) is 100˚. Direct connection of the CS-mount objective to the cooled CCD is not possible and, therefore, a Thorlabs re-imager optics is needed. The objective is placed at 45˚ from the direction of the beams with the pinhole being*

*placed between both beams as depicted in Fig. S1. Based on the geometry (Fig. S1), the pinhole together with a given laser beam define a scattering plane. […]"*

L162-164: Depending on the CPC mode used, this would imply that the aerosol flow to the uNeph was around 3.5 or 4.7 L min$^{-1}$ – can you clarify what you estimate the volume flow in the uNeph to be (and hence the estimated residence time).

During the test the CPC flow rate was set to 0.3 lpm. This implies that the flow rate through the instrument was ~ 4.7 lpm, which translates to a lower limit residence time of ~30 s given a chamber volume of ~2.5 l and with assuming plug flow.

L485; L721-22: I believe the N limit in the summations over i indices refers to the number of angles – is that correct? It's unclear whether this is incremented by pixel angle or by absolute angle. How does the angular resolution of measurements vary across the full range of measurement?

Yes, $N$ corresponds to the number of extracted angles, which was done on a regular grid with 1° angular resolution. The symbol $N$ is now defined in the revised text.

Technically, the maximal possible resolution is related to the field of view of the illuminated pixels. We did not attempt a precise assessment of the maximally possible resolution as a function of θ, because effective information content is often limited by other factors, such as random noise. If the field of view per pixel was substantially greater than 1°, then this would cause smearing of the phase function extracted at 1° resolution compared to the actual phase function. The results for narrow monodisperse aerosol samples do not indicate substantial smearing of the signal.

Section 3.1.3 was updated to include some of these considerations:
*"The blue lines in Figs. 2b and 2c illustrate the beam cross-section at θ=40° before and after transformation as an example. Section A4 provides a more detailed description of the image transformation. A regular grid with 1° angular resolution was chosen for the extracted image. No attempts were made to extract the signal with higher angular resolution, given that the information content of measured phase functions is often limited by measurement uncertainties rather than angular resolution."*

L592: It would appear that the V$_{tot}$ predicted by the retrieval systematically overestimate the volume when compared to the SMPS measurement. Could this be because the measured size distribution (Figure 8) appears by eye to deviate from an ideal lognormal distribution?

We doubt that a deviation from log normal size distribution shape is the main reason for the systematic difference between SMPS measurement and uNeph retrievals. A similar difference was observed using the binned size distribution variant for the retrieval, which is not constrained to a lognormal shape. Therefore, we retain the statement made in Section 5.3: *"The reasons for this discrepancy remain elusive."* Having said so, ongoing work shows that better agreement can be achieved, possibly as a result of improved independent measurements of aerosol volume concentration.

L618-619: It would be interesting and useful to test your calibration and data processing procedures for absorbing spherical particles in the future.

Indeed, this is part of our follow-up work.

L797: This seems like a very conservative (i.e. erring on the side of large uncertainty) way of quantifying this error. No corrections needed, just an observation!

Table 1: Please clarify whether "q25" and "q75" refer to the 25[th] and 75[th] percentile quantile points, or other values.

We changed the notation and the caption of Table 1:
*"Q1 and Q3 refer to the lower and upper quartiles, respectively."*

Figure 2: No image appears in the manuscript

This looks like an unfortunate compatibility issue. The preprint published on AMT's web size does not suffer from this issue, at least not when opened on our computers.

Figure 5: It is unclear the purpose for showing the line depicting "3% of uNeph air measurement" in the lefthand panels

The caption of previous Fig. 5 (now Fig. 7) has been updated:
*"[…]The contribution of air background uncertainty to measurement error remains small (magenta lines in panels to the right) unless the aerosol phase function (red line in panels to the left) approaches values only slightly above 3% of the air background (blue dashed lines). This effect is nicely seen in panels (b) and (f) at angles ~105° and ~155°.[…]"*

Figure 6: Check units on stated number concentrations in caption

We corrected the number concentration units to cm$^{-3}$ for all the instances in the caption of Fig. 8 (formerly Fig. 6).

Figure 8: The information presented in (b,c) plots might be more clearly depicted in a table

We agree that this figure can be improved. To do so, we have modified it to present the refractive index as a complex number. New Figure 10 (formerly Fig. 8):

[Figure]

General comments on Figures 6, 7, S2, S4, S8, S9, S11, S12, S13, S14, S15, S16, and S17 :
The font size in the axes labels, legends, and/or schematics is far too small.

We have updated all the aforementioned figures and increased the font size such that the texts in the figures are more legible.

**Reviewer #2:**

The manuscript provides a detailed explanation of a new imaging nephelometer called µNeph. Data processing methods are described, followed by validation of the measurements with monodisperse and monomodal polydisperse test aerosols and then finally an example of the potential to perform retrievals of particle size and concentration using µNeph data. The demonstrated accuracy of the µNeph is impressive, especially considering its small size, and I expect it to contribute significantly to our ability to measure aerosol phase matrix elements. Additionally, some aspects of the described data reduction procedure represent advancements over prior approaches, like the use of a monatomic gas (e.g., argon) to characterize the stokes vector of the incoming laser light source. The manuscript is clear and well written making it a very useful resource for future users of µNeph data and other imaging nephelometer efforts. I can thus confidently recommend publication, after the following minor points have been addressed.

SPECIFIC COMMENTS

LN 30: I think this should read "...aerosols makes them difficult to..."

We changed this sentence.

LN 36: It should just be "radiance" here, not "irradiance".

Correct. We changed the text accordingly.

LN 71: It may be good to also cite the very well known Amsterdam–Granada polar nephelometer that is capable of measuring all phase matrix elements (e.g., Muñoz et al. 2012).

The instrument by Muñoz et al. (2012) is now mentioned in the text.

LN 95: Would the authors be able to elaborate on potential uses that they envision for the instrument outside of the laboratory (e.g., perhaps onboard a UAV)? The small size of the instrument seems like it would be advantageous for many such applications.

The conclusions have been updated:
*"[…]Overall, the uNeph was successfully validated and our results shows the applicability of a down sized laser imaging nephelometer. Downsizing the instrument posed challenges which result in slower measurements of aerosol samples containing large particles present in small number. At the same time, dimensions of the uNeph have be considerably reduced compared to previous laser imaging nephelometers. This demonstrates that operation of such instruments on e.g. unmanned aerial vehicles is achievable. Such increased flexibility enables acquisition of in situ aerosol polarimetry data sets, which can greatly benefit the remote sensing community for validating and improving existing retrieval algorithms. Future improvements to the uNeph will aim at increasing the dynamic range and the number of measurement wavelengths, with the goal to use it in laboratory and field settings for aerosol characterization, as well as for validation and optimization of polarimetric aerosol property retrievals for more complex aerosols. […]"*

LN 116: Perhaps this would be clearer as greater/less than symbols (e.g., "0≤θ≤90°") to avoid confusion with the minus operator.

We changed the text according to the reviewer's recommendation.

LN 117: Presuming the laser beams fill most of the length of the image, this CCD resolution would imply a raw measurement resolution of closer to 0.1°. Are multiple beam cross section integrals averaged to obtain the coarser 1° resolution? Also, how was 1° selected as the final resolution?

Yes, CCD resolution is higher than 1° in θ. However, signal was extracted on a fixed grid of 1° angular resolution and this was done by selecting a subset of pixels rather than grouping multiple pixels. The choice of 1° angular resolution was a subjective choice. Technically, the maximal possible resolution is related to the field of view of the illuminated pixels. We did not attempt a precise assessment of the maximally possible resolution as a function of θ, because effective information content is often limited by other factors, such as random noise. If the field of view per pixel was substantially greater than 1°, then this would cause smearing of the phase function extracted at 1° resolution compared to the actual phase function. The results for narrow monodisperse aerosol samples do not indicate substantial smearing of the signal. (See also response to related questions by reviewer 1.

Section A4 was slightly adapted to clarify transformation of the image to angle-pixel coordinates:
*"Sections 3.1.2 and A3 described how to obtain the red fit curve in Fig. 2b which is a parametrization of pixels coordinates as a function of θ along the centre line of the beam on the CCD image. The next goal is to extract a pixel array representing laser beam image cross-sections for each polar angle ($\theta_i$). For this purpose, lines perpendicular to this fit were obtained as:*

$$y - y_{CL}(\theta_i) = -\frac{1}{m(\theta_i)}(x - x_{CL}(\theta_i)) \hspace{3cm} (A4)$$

*In Eq. A4, $y_{CL}(\theta_i)$ and $x_{CL}(\theta_i)$ are the x and y coordinates of points along the laser beam centre line and $m(\theta_i)$ is the gradient of the red fit curve at angle $\theta_i$. The angles $\theta_i$ were chosen to represent a regular angular resolution of 1°. We then considered the pixels closest to these perpendicular lines to extract the beam cross sections for each ($\theta_i$). Pixels in between these perpendicular lines were ignored. Figure S7 shows the transformed image with beam cross sections as a function of θ. 120 pixels were extracted for each angle such that the full beam cross-section is included in the transformed image."*

LN 118: I assume "A/D" stands for "analog-to-digital" but it still may be good to define the abbreviation explicitly.

We changed the text according to the reviewer's recommendation.

LN 121: Please specify the diameter of the pinhole used to image the lasers.

Added in Section 2.1:
*"[...]One design element of the uNeph is the use of a wide field of view pinhole lens in the camera objective, which enables the instrument to be downsized. The camera objective is the Marshall V-PL25CS-12 (discontinued) with a pinhole size of ~ 2 mm, a focal length of 2.5mm*

*and aperture of F2.8. The field of view (diameter) is 100˚. Direct connection of the CS-mount objective to the cooled CCD is not possible and, therefore, a Thorlabs re-imager optics is needed. The objective is placed at 45˚ from the direction of the beams with the pinhole being placed between both beams as depicted in Fig. S1. Based on the geometry (Fig. S1), the pinhole together with a given laser beam define a scattering plane. [...]"*

LN 127: It would be helpful to state exactly where the Thorlabs PAX polarimeter was placed in the optical path. Is it possible to place it in the measurement chamber? If so, why are q1 and q2 not known exactly, but instead later solved for using argon measurements? Perhaps this is due to the lack of exact knowledge of the camera's pinhole in the Thorlabs polarimeter's coordinate system? Please elaborate.

We revised the manuscript as follows:
*"A polarimeter (PAX1000VIS/M, Thorlabs Inc.) was used to verify that the two laser beams crossing the chamber are almost fully linearly polarized. Uncertainties in pinhole alignment relative to the laser beams is estimated to have larger effect on deviations from nominal linear polarization than the quality of polarization control."*

See also answers to other comments addressing q1 and q2 for further detail on this aspect.

LN 180: Apologies if I missed it, but how frequently are dark images obtained? Are these coefficients updated during a given measurement?

Dark current correction parameters were determined once and held fixed for all analyses. Dark signal interference only becomes relevant when aggregating many images with very short exposure times to avoid saturation in the presence of large particles. It might be possible to push detection limits a little further with more precise dark signal correction. However, we envisage other modifications to circumvent need for very short exposure times.

LN 194: It would be helpful to replace "several" with the exact number of pinhead positions.

The updated text is:
*"For this purpose, a pinhead mounted on a 3D translation stage is placed at 27 different positions inside the forward and the backward laser beams (Figs. S5 and S6)."*

LN 234: While some referencing of supplement figures in the main text in passing is certainly fine, here and in several other places (e.g. LN 258, LN 329, LN 390, LN 499, etc.), a figure from the supplement is described in very significant detail. In these cases where whole sentences or paragraphs are devoted to supplement figures I would suggest either moving that text to the supplement, or the figure into the main text.

We considered this suggestion as follows: Former Figures S12 and S13 were merged into one and placed it in the main text (now Fig. 5). We further included the former Figure S20 in the main text (now Fig. 6). Discussion of the former Fig. S24 (now Fig. S21) was replaced by a brief interpretation of this figure (see Section 5.2), while the more complete discussion was moved to the figure caption.

LN 240: Are these $\Xi$ limits fixed or a function of scattering angle?

Yes, the limits vary with angles as depicted in Figure S10.

LN 240: Will the value of Ξ at which saturation occurs be dependent on aerosol loading and dark current? The effect is probably small, but it seems that short exposures (low dark current) and high loading (high counts in center of last beam) would be more likely to saturate pixels at the center of the beam than a longer exposure with lower aerosol loading, even if both cases produced the same value of Ξ.

Yes, a very small effect is expected as dark signal contribution already is subtracted from Ξ, while saturation occurs at a fixed level. Therefore, we only use Ξ for filtering short exposure times with too low signal, whereas long exposure times are filtered by identification of actually saturated pixels. Nevertheless, the high limit for Ξ remains useful to indicate the approximate dynamic range.

LN 282: Have the authors considered using a beam expander? This was considered for the PI-Neph of Dolgos and Martins (2014) in order to reduce the signal at each individual pixel, while also enlarging the sampled volume providing statistics for a larger number of particles in each image acquisition.

Using a beam expander would indeed help with enlarging the sample volume. We did not follow this pathway for stray light mitigation reasons. Furthermore, enlarging the beam could introduce relevant variations of azimuthal scattering angle η across the beam cross section.

LN 287: Is the precision of the laser reference photodetector known? If so, it would be helpful to state it.

We did not assess whether the small instrument drift report in the stability tests is related to insufficient precision of the photodetector. The model number is now included in the technical description.

LN 328: Do the authors have any theories as to the primary drivers of the instrument calibration drift?

We were unable to identify the causes for such drifts in calibration factors. These are possibly related to temperature effects as the instrument does not exhibit drift on longer time scales going beyond this short-term variability.

Eq 3: I'm wondering if something like $F\parallel$ and $F\perp$ would be clearer than F1 and F2, which are very easily confused with phase matrix elements. Although, perhaps the close connection (and same units) with phase matrix elements make the variable names appropriate.

In our manuscript we reserved the $\parallel$ and $\perp$ symbols for fully parallel and perpendicular polarization states with respect to the scattering plane. The reason that we chose $F_1$ and $F_2$ for the measurements at the two polarization setpoints is to emphasize that these date are not (necessarily) reflecting light scattering for fully parallel and perpendicular polarization states.

LN 371: Is there a physical basis for the assumption |q1|=|q2|? (Or, more precisely, I think q1=-q2, right?) Are polarization artifacts expected to impact q1 and q2 identically in some way?

This is an approximation that falls within the uncertainty of q1 and -q2. Some effects are expected to affect q1 and -q2 in the same way, e.g. the impacts of imperfect geometry or the degree of linear polarization achieved by the polarizer. Perpendicular orientation of the two

polarization setpoints is expected to be achieved with high accuracy, such that error in these setpoints should not introduce large difference between q1 and -q2.

LN 384: Please clarify if 1.7% is the width of the PSL size distribution or uncertainty on the mean PSL diameter. If the former, was an uncertainty on mean diameter provided by the PSL manufacturer? If so, how much variation does that uncertainty translate to in terms of Mie calculated F11 and F12?

The 1.7% is the coefficient of variation (CV) referring to the size distribution width of the PSL size standard. A CV value of 1.7% translates to a GSD of ~ 1.017. The PSL distributor also provided the uncertainty of the reported mean diameter, which was ±9 nm for the 600 nm PSL sample. We modified Figure 6 to include the corresponding propagated uncertainty for the predicted phase functions, which is relevant for part of the backscatter range. When using PSL size standards for adjusting the angle calibration, it is important to validate the calibration using other test aerosol to minimize the risk of blending calibration aerosol bias into the angle calibration.

Additionally, we noticed typos of the values listed in Table 1: Erroneous values 0.07 and 0.05 have been corrected to read 1.7% and 1.5%, respectively. This are the actual CV values reported by the PSL distributer.

Sec 4.3: Would it be possible to further refine q, and characterize q1 and q2 separately, using the maxima and minima of the PSL measurements of F1 and F2?

This could possibly be done. However, the combined effects of angular uncertainties and q uncertainties could make it difficult to properly isolate and characterize $q_1$ and $q_2$ values individually such that this effort pays off with increased accuracy.

Eq 4: Is instrument stray-light background implicitly included here or is it not considered in the error model?

Yes, it is implicitly included in the error of background subtraction. If the stray light error at a certain angle was exceeding the allowance for background subtraction error, then the resulting measurement error would become larger than estimated.

LN 411: For consistency, maybe drop the squaring "2" on "σ^2_e,l" or add it to the other three variables.

We changed the text according to the reviewer's recommendation.

Figure 5: I'm not see a section 3.5.3, as referenced in the caption. Please update to the correct section number.

Cross-reference has been corrected in the figure caption (now Figure 7).

LN 463: What was the reason the values 1.04 and 1.08 were assumed?

The GSD of the idealized and perfect triangular transfer function without considering diffusion at an AAC resolution parameter Rs=20 (our operating condition) is as small as ~1.021. This would have been an extremely conservative low limit for GSD. In reality the resolution was

likely somewhat poorer than the ideal value, hence GSD=1.04 seems justified as lower limit. SMPS measurements of AAC selected samples with comparable settings provided a measured GSD of ~1.08. This value is taken as an upper limit given the GSD measured by the SMPS is a convolution of PSL size distribution GSD and DMA transfer function width. While these limits are somewhat subjective, they should provide a reasonable estimate of uncertainty in calculated phase functions associated with GSD uncertainty.

LN 493: This is the second time the blue lines are defined as best fits.

We adjusted the text to avoid repetition of the best fit definition.

LN 534: Please explain how GRASP was tailored to produce the µNeph-GRASP Inversion. What changes were made relative to the standard build of GRASP? Also, for future work, it may be helpful to note that there was a high size parameter resolution version of the GRASP kernels developed to retrieve PSL properties in Espinosa et al. (2017). Those files are likely available by request from the GRASP team.

We use the standard build of GRASP-OPEN for these runs without any modifications in the source code. Limited resolution of the standard GRASP kernels with respect to diameter is not an issue in the example presented here. However, it is good to know that the refined kernels exist, as discretization errors may become important for property retrievals of quasi-monodisperse aerosol samples.

LN 549: I think "F11/F12" should be "F12/F11".

This typo has been corrected.

REFERENCES

Muñoz, O., Moreno, F., Guirado, D., Dabrowska, D. D., Volten, H., & Hovenier, J. W. (2012). The Amsterdam–Granada light scattering database. Journal of Quantitative Spectroscopy and Radiative Transfer, 113(7), 565-574.